# Offline Reinforcement Learning with Additional Covering Distributions

**Chenjie Mao** *chenjiemao@hust.edu.cn*
*School of Computer Science and Technology*
*Huazhong University of Science and Technology*

**Reviewed on OpenReview:** *https://openreview.net/forum?id=AfXq3x3X16*

## Abstract

We study learning optimal policies from a logged dataset, i.e., offline RL, with function general approximation. Despite the efforts devoted, existing algorithms with theoretic finite-sample guarantees typically assume exploratory data coverage or strong realizable function classes (e.g., Bellman-completeness), which is hard to be satisfied in reality. While there are recent works that successfully tackle these strong assumptions, they either require the gap assumptions that could only be satisfied by part of MDPs or use the behavior regularization that needs to be carefully controlled. To solve this challenge, we provide finite-sample guarantees for a simple algorithm based on marginalized importance sampling (MIS), showing that sample-efficient offline RL for general MDPs is possible with only a *partial coverage* dataset (instead of assuming a dataset covering all possible policies) and *weak realizable* function classes (assuming function classes containing simply one function) given additional side information of a covering distribution. We demonstrate that the covering distribution trades off prior knowledge of the optimal trajectories against the coverage requirement of the dataset, revealing the effect of this inductive bias in the learning processes. Furthermore, when considering the exploratory dataset, our analysis shows that only realizable function classes are enough for learning near-optimal policies, even with no side information on the additional coverage distributions.

## 1 Introduction and related works

In offline reinforcement learning (offline RL, also known as batch RL), the learner tries to find good policies with a pre-collected dataset. This data-driven paradigm eliminates the heavy burden of environmental interaction required in online learning, which could be dangerous or costly (e.g., in robotics (Kalashnikov et al., 2018; Sinha & Garg, 2021) and healthcare (Gottesman et al., 2018; 2019; Tang et al., 2023)), making offline RL a promising approach in real-world applications.

In early theoretic studies of offline RL (e.g., Munos (2003; 2005; 2007); Ernst et al. (2005); Antos et al. (2007); Munos & Szepesvari (2008); Farahmand et al. (2010)), researchers analyzed the finite-sample behavior of algorithms under the assumptions such as *exploratory* datasets, realizable or Bellman-complete function classes. However, despite some error propagation bounds and sample complexity guarantees achieved in these works, the strong coverage assumption made on datasets and the *non-monotonic* assumptions made on function classes—which are always hard to be satisfied in reality—drive people to try to find sample-efficient offline RL algorithms under only weak assumptions about dataset and function classes (Chen & Jiang, 2019).

From the *dataset perspective*, partial coverage, which means that only some specific (or even none) policies are covered by the dataset (Rashidinejad et al., 2021; Xie et al., 2021; Uehara & Sun, 2021; Song et al., 2022), is studied. To address the problem of insufficient information, most algorithms use *behavior regularization* (e.g., Laroche & Trichelair (2017); Kumar et al. (2019); Zhan et al. (2022)) or *pessimism in the face of uncertainty* (e.g., Liu et al. (2020); Jin et al. (2020); Rashidinejad et al. (2021); Xie et al. (2021); Uehara & Sun (2021); Cheng et al. (2022); Zhu et al. (2023)) to constrain the learned policies to be close to the

behavior policy. Most of the algorithms in this setting (except some that we will discuss later) require function assumptions in some sense of *completeness*—Bellman-completeness or strict realization according to another function class (we attribute it as strong realization).

From the *function classes perspective*, while the primary concern is Bellman-completeness assumption which is criticized for its non-monotonicity, some recent works (Zhan et al., 2022; Chen & Jiang, 2022; Ozdaglar et al., 2022) have noticed that the realizability according to another function class is also non-monotonic. These non-monotonic properties contradict the intuition in supervised learning that rich function classes perform better (or at least no worse). Typical examples of these assumptions are the "realizability of all candidate policies' value functions" (e.g., Jiang & Huang (2020); Zhu et al. (2023)) and the "realizability of all candidate policies' density ratio" (e.g., Xie & Jiang (2020b)). These assumptions are equally strong as Bellman-completeness, and we classify them as "strong realizability" (Zhan et al. (2022); Ozdaglar et al. (2022) attribute it as "completeness-type") for clarification. We also classify assuming that the function class realizes specific elements as "weak realizability" correspondingly (Chen & Jiang (2022) attributes this as "realizability-type"). We argue that this taxonomy is justified also because Bellman-completeness can be converted to the realizability assumption between two function classes with the minimax algorithm (Chen & Jiang, 2019). This conversion aligns the behavior of Bellman-completeness with strong realizability assumptions.

On the one hand, Bellman-completeness assumption is always made in the classical finite-sample analyses of offline RL (e.g., analysis of FQI (Ernst et al., 2005; Antos et al., 2007)) to ensure closed updates of value functions (Sutton & Barto, 2018; Wang et al., 2021). This assumption is notoriously hard to mitigate, and Foster et al. (2021) even suggests an information-theoretic lower bound stating that without Bellman-completeness, sample-efficient offline RL is impossible even with an exploratory dataset and a function class containing all candidate policies' value functions. Therefore, it is clear that additional assumptions are required to circumvent Bellman-completeness.

On the other hand, as marginalized importance sampling (MIS, see, e.g., Liu et al. (2018); Uehara et al. (2019); Jiang & Huang (2020); Huang & Jiang (2022)) has shown its effect of eliminating Bellman-completeness with only a partial coverage dataset by assuming the realizability of density ratios in off-policy evaluation (OPE), there are works try to adapt it to policy optimization. These adaptations retain the elimination of Bellman-completeness, but most come up with other drawbacks. Some works (e.g., Jiang & Huang (2020); Zhu et al. (2023)) use OPE as an intermediate evaluation step for policy optimization yet require the strong realizability assumption on the value function class. The others borrow the idea of discriminators from MIS. Lee et al. (2021); Zhan et al. (2022) take value functions as discriminators for the optimal density ratio, using MIS to approximate the linear programming approach of Markov Decision Processes (Manne, 1960; Puterman, 1994). Nachum et al. (2019); Chen & Jiang (2022); Uehara et al. (2023) take distribution density ratios as discriminators for optimal value function by replacing the Bellman equation in OPE with its optimality variant. While in most cases, theoretic finite-sample guarantees with these discriminators would require strong realizable function classes (e.g., Rashidinejad et al. (2022)), Zhan et al. (2022); Chen & Jiang (2022); Uehara et al. (2023) avoid this with additional gap assumptions or an alternative criterion of optimality—performance degradation w.r.t. the regularized optimal policy. To the best of our knowledge, they are the only works that achieve theoretic sample-efficient guarantees under only weak realizability and partial coverage assumptions. However, on the one hand, the gap (margin) assumption (Chen & Jiang, 2022; Uehara et al., 2023) causes that only some specific Markov decision processes (MDPs)—under which the optimal value functions have gaps—can be solved. On the other hand, sub-optimality compared with a regularized optimal policy (Zhan et al., 2022) could be meaningless in some cases, and the actual performance of the learned policy requires a careful fine-tuning on the regularization. Moreover, the sample complexities from Uehara et al. (2023) are polynomial w.r.t. the cardinality of the action space, which cannot handle large action spaces.

As summarized above, the following question arises:

> *Is sample-efficient offline RL possible with only partial coverage and weak realizability assumptions for general MDPs?*

Table 1: A comparison of offline RL algorithms without assuming Bellman-completeness and model-realizability. "conc." stands for concentrability. The data assumptions from Xie & Jiang (2020a) require that for all $s \in \mathcal{S}$, $a \in \mathcal{A}$, and $s' \in \mathcal{S}$, we have that $d^{\mathcal{D}}(s,a)$, $1/\pi_b(a|s)$, $P(s'|s,a)/\mu^{\mathcal{D}}(s')$ and $\mu^{\mathcal{D}}(s)/\mu_0(s)$ are lower bounded by a constant (we define $0/0 := 1$). $\mathcal{C}(\mathcal{Q})$ stands for the convex hull of $\mathcal{Q}$. $w_\alpha^\star$ and $v_\alpha^\star$ are the regularized optima from the original paper. $\mathcal{U}$ and $\mathcal{Z}$ are the newly defined function classes introduced by Rashidinejad et al. (2022). Uehara et al. (2023) is compromised by two parts for algorithms with or without regularizations. The unregularized algorithm part requires margin assumptions of the optimal value function, and the sample complexities of both parts are polynomial w.r.t. $|\mathcal{A}|$. $d_c$ is the additional covering distribution introduced by this paper. The definitions of the other notations can be found in the remaining part of this paper.

| Algorithm | Data assumptions | Function assumptions | Drawbacks |
|---|---|---|---|
| Jiang & Huang (2020) | optimal conc. | $w^\star \in \mathcal{W}$, and $\forall \pi \in \Pi, Q_\pi \in \mathcal{C}(\mathcal{Q})$ | strong realizability |
| Xie & Jiang (2020a) | conc. w.r.t. $P$ and $\mathcal{A}$ | $Q^\star \in \mathcal{Q}$ | strong dataset assumptions |
| Zhan et al. (2022) | optimal conc. | $w_\alpha^\star \in \mathcal{W}$, and $v_\alpha^\star \in \mathcal{V}$ | Use regularization |
| Chen & Jiang (2022) | optimal conc. | $w^\star \in \mathcal{W}$, and $Q^\star \in \mathcal{Q}$ | assume gap (margin) |
| Rashidinejad et al. (2022) | optimal conc. | $w^\star \in \mathcal{W}, V^\star \in \mathcal{V}, u_w^\star \in \mathcal{U} \ \forall w$ and $\zeta_{w^\star,u}^\star \in \mathcal{Z} \ \forall u$ | strong realizability |
| Zhu et al. (2023) | optimal conc. | $w^\star \in \mathcal{W}$, and $\forall \pi \in \Pi, Q_\pi \in \mathcal{Q}$ | strong realizability |
| Uehara et al. (2023) | optimal conc from $d^{\mathcal{D}}$ | $w^\star \in \mathcal{W}$, and $Q^\star \in \mathcal{Q}$ | assume gap (margin)/$|\mathcal{A}|$ |
| Ours (VOPR) | optimal conc. from $d_c$ | $w^\star \in \mathcal{W}$, $\beta^\star \in \mathcal{B}$ and $Q^\star \in \mathcal{Q}$ | assume a covering $d_c$ |

We answer this question in the positive and propose an algorithm that achieves finite-sample guarantees under weak assumptions with the help of an additional covering distribution. We assume that the covering distribution covers all non-stationary near-optimal policies, and the dataset covers the trajectories induced by an optimal policy from it. The covering distribution can be taken as a minimum requirement for accurately estimating the optimal value function $Q^\star$, and we use it as a regularizer in our algorithm. It is *adaptive* such that both "non-stationary" and "near-optimal" above would be alleviated as the gap of optimal value function increases. The covering distribution also gives a trade-off against the data coverage assumption: the more accurate it is, the fewer redundant trajectories are required to be covered by the dataset. Furthermore, we can directly use the data distribution as the covering distribution as done in Uehara et al. (2023), if the near-optimal variant of their data assumptions are also satisfied.

For comparison, we summarize algorithms that do not need Bellman-completeness and *model realizability* (which is even stronger (Chen & Jiang, 2019; Zhan et al., 2022)) in Table 1. Necessary transfers are made to get the sub-optimality bound.

In conclusion, our contributions are as follows:

- (Section 3) We identify the novel mechanism of non-stationary near-optimal concentrability in policy optimization under weak assumptions.

- (Section 4) We demonstrate the trade-off brought by additional covering distributions for the coverage requirement of the dataset.

- (Section 4) We propose the first algorithm that achieves finite-sample guarantees for general MDPs under only weak realizability and partial coverage assumptions.

## 2 Preliminaries

This section introduces base concepts and notations in offline RL with function approximation and MIS. See Table 2 in Appendix A for a more complete list of definitions of notations.

**Markov Decision Processes (MDPs)** We consider infinite-horizon discounted MDPs defined as $(\mathcal{S}, \mathcal{A}, P, R, \gamma, \mu_0)$, where $\mathcal{S}$ is the state space, $\mathcal{A}$ is the action space, $P\colon \mathcal{S} \times \mathcal{A} \to \Delta(\mathcal{S})$ is the transition probability, $R\colon \mathcal{S} \times \mathcal{A} \to [0, R_{max}]$ is the *deterministic* reward function, $\gamma \in (0,1)$ is the discount factor that unravels the problem of infinite horizons, and $\mu_0 \in \Delta(\mathcal{S})$ is the initial state distribution. With a policy $\pi\colon \mathcal{S} \to \Delta(\mathcal{A})$, we say that it induces a random trajectory $\{s_0, a_0, r_0, s_1, a_1, r_1, \ldots, s_i, a_i, r_i, s_{i+1}, \ldots\}$

if: $s_0 \sim \mu_0$, $a_i \sim \pi(\cdot|s_i)$, $r_i = R(s_i, a_i)$ and $s_{i+1} \sim P(\cdot|s_i, a_i)$. We define the expected return of a policy $\pi$ as $J_\pi = \mathbb{E}\left[\sum_{i=0}^\infty \gamma^i r_i \mid \mu_0, \pi\right]$. We also denote the value function of $\pi$ as the expected return starting from some specific state $s$ or state-action pair $(s,a)$ as $V_\pi(s) = \mathbb{E}\left[\sum_{i=0}^\infty \gamma^i r_i \mid s, \pi\right]$ and $Q_\pi(s, a) = \mathbb{E}\left[\sum_{i=0}^\infty \gamma^i r_i \mid (s, a), \pi\right]$. We denote the optimal policies that achieve the maximum return $J^\star$ from $\mu_0$ as $\Pi^\star$, and its member as $\pi^\star$. We say a policy is optimal almost everywhere if its state value function is maximized almost everywhere and denote it as $\pi_e^\star$ ($\pi_e^\star$ is not always unique). We represent the value functions of $\pi_e^\star$ as $V^\star$ and $Q^\star$. It worth noting that $V^\star$ and $Q^\star$, the *unique* solutions of both Bellman optimality equation and the primal part of LP approach of MDPs (Puterman, 1994), are *not* the value functions of all optimal policies. For ease of discussion, we assume $\mathcal{S}$, $\mathcal{A}$, $\mathcal{S} \times \mathcal{A}$ are compact measurable spaces and, with abuse of notation, we use $\nu$ to denote the corresponding finite uniform measure on each space (e.g., Lebesgue measure). We use $P_\pi$ to denote the state-action transition operator for density $d$ as $P_\pi d(s', a') \coloneqq \int_{\mathcal{S} \times \mathcal{A}} \pi(a' \mid s') P(s' \mid s, a) d(s, a) d\nu(s, a)$. The induced distribution of a policy $\pi$ is defined as $d_\pi(s, a) = (1 - \gamma) \sum_0^\infty \mathbb{P}(s_i = s, a_i = a | s_0 \sim \mu_0, a_i \sim \pi(\cdot|s_i))$, and use $\mu_\pi$ to denote the state margin of $d_\pi$. We would add subscripts to denote distributions not induced from $\mu_0$ (e.g., $d_{d', \pi}$).

**Offline policy learning with function approximation** In the standard theoretical setup of offline RL, we are given with a dataset $\mathcal{D}$ consisting of $N$ $(s, a, r, s')$ tuples, which is collected with *state* distribution $\mu^D$ and behavior policy $\pi_b$ such that $s \sim \mu^D, a \sim \pi_b(\cdot|s), r = R(s, a), s' \sim P(\cdot|s, a)$. We use $d^{\mathcal{D}}(s, a) \coloneqq \mu(s)\pi_b(a \mid s)$ to denote the composed state-action distribution of the dataset. When the state space and action space become rather complex, function approximation is typically used. For this, we assume there are some function classes at hand that satisfy certain assumptions and have limited complexity (measured by cardinality, metric entropy and so forth). The function classes considered in this paper are state-action value function class $\mathcal{Q} \subseteq (\mathcal{S} \times \mathcal{A} \to \mathbb{R})$, state distribution ratio class $\mathcal{W} \subseteq (\mathcal{S} \times \mathcal{A} \to \mathbb{R})$, and policy ratio class $\mathcal{B} \subseteq (\mathcal{S} \times \mathcal{A} \to \mathbb{R})$.

**MIS with density discriminators and $L^2$ error bound** One of the most popular ways to estimate the optimal value function is via the Bellman optimality equation:

$$\forall s \in \mathcal{S}, a \in \mathcal{A}, \quad Q^\star(s, a) = T^\star Q^\star(s, a) \tag{1}$$

where $T^\star q(s, a) \coloneqq R(s, a) + \gamma \mathbb{E}_{s' \sim P(\cdot|s, a)}[\max q(s', \cdot)]$ denotes the Bellman optimality operator. However, when we try to utilize the constraints from Eq. (1) (e.g., through the $L^1$ error $\|q - T^\star q\|_{1, d^{\mathcal{D}}}$[1]), the expectation in $T^\star$ would introduce the infamous double-sampling issue (Baird, 1995), making the estimation intractable.

To overcome this, privious works with MIS tried to take weight functions as discriminators and minimize a weighted sum of Eq. (1). In fact, even the $L^1$ error itself could be written as a weighted sum with some sign function (take 1 if $q \geq T^\star q$ and $-1$ otherwise (Ozdaglar et al., 2022)). Namely, we approximate $Q^\star$ through

$$\hat{q} = \underset{q \in \mathcal{Q}}{\operatorname{argmin}} \max_{w \in \mathcal{W}} \mathbb{E}_{d^{\mathcal{D}}}[w(s, a)(q(s, a) - T^\star q(s, a)]. \tag{2}$$

Since the weight function class $\mathcal{W}$ is marginalized into the state-action space (instead of trajectories), this approach is called marginalized importance sampling (MIS) (Liu et al., 2018). While theoretic guarantees in MIS under weak realizability and partial coverage assumptions are typically made for scalar values (e.g., the return (Uehara et al., 2019; Jiang & Huang, 2020)), recently, Zhan et al. (2022); Huang & Jiang (2022); Uehara et al. (2023) have gone beyond this and derived $L^2$ error guarantees for the estimators by using some strongly convex functions. Among them, the optimal value function estimator from Uehara et al. (2023) constructs the base of this work.

## 3 From $Q^\star$ to optimal policy, the minimum requirement

Uehara et al. (2023) shows that accurately estimating optimal value function $Q^\star$ under $d^{\mathcal{D}}$ is possible if $d^{\mathcal{D}}$ covers the optimal trajectories starting from itself. This "self-covering" assumption could be relieved and

---

[1] $\|x\|_{p,q}$ denotes $q$ weighted $L^p$ norm, i.e., $\|x\|_{p,q} = \left(\int x^p dq\right)^{1/p}$.

generalized if we only require an accurate estimator under some state-action distribution $d_c$, such that $d_c$ is absolutely continuous w.r.t. $d^{\mathcal{D}}$, i.e., $d_c \ll d^{\mathcal{D}}$ (we use $\mu_c$ and $\pi_c$ to denote the state distribution and behaviour policy decomposed from $d_c$). In fact, $d_c$ provides a trade-off for the coverage requirement of the dataset: the fewer state-action pairs on the support of $d_c$, the weaker data coverage assumptions we will make. Nevertheless, how much trade-off can $d_c$ provide while preserving the desired result?

In policy learning, our goal is to derive an optimal policy $\hat{\pi}$ from the estimated $Q^\star$ (denoted as $\hat{q}$). While there are methods (see Section 4.3 for a brief discussion) that induce policies from $\hat{q}$ by exploiting pessimism or data regularization, one of the most straightforward ways is to take the actions covered by $d_c$ that achieve the maximum $\hat{q}$ in each state. This can be done with the help of policy ratio class $\mathcal{B}$, via

$$\hat{\beta} = \operatorname*{argmax}_{\beta \in \mathcal{B}} \langle \mu_c, \hat{q}(\cdot, \pi_\beta) \rangle \quad \text{and take} \quad \hat{\pi} = \pi_{\hat{\beta}}, \tag{3}$$

where $\pi_\beta(a \mid s) = \pi_b(a \mid s)\beta(s, a)$ (normalized if needed). In case of infinite amount of data, we have $q^\star(s, a) = Q^\star(s, a)$ on the support of $d_c$. With the optimal realizability of $\mathcal{B}$ and concentrability of $\pi_c$, Eq. (3) is actually equivalent to

$$\langle \mu_c, Q^\star(\cdot, \hat{\pi}) - Q^\star(\cdot, \pi_e^\star) \rangle = 0, \tag{4}$$

which guides us to exploit the coverage provided by $\mu_c$. Recall that our goal is to use $d_c$ to trade off the coverage assumption of $d^{\mathcal{D}}$. Therefore, the question left, which forms the primary subject of this section, is

*With which $\mu_c$ can we conclude that $\hat{\pi}$ is optimal from $\langle \mu_c, Q^\star(\cdot, \hat{\pi}) - Q^\star(\cdot, \pi_e^\star) \rangle = 0$,*
*and what is the minimum requirement of it?*

Since $\mu_c$ and $d_c$ are to provide additional coverage, we also call them "covering distributions".

The remainder of this section is organized as follows: we first show why single optimal concentrability of $\mu_c$ is not enough in Section 3.1, and then we introduce the alternative "all optimal concentrability" in Section 3.2 and the adapted version of it in Section 3.3 to accommodate statistical errors.

### 3.1 The dilemma of single optimal contentrability

Single optimal concentrability is standard (Liu et al., 2020; Xie et al., 2021; Cheng et al., 2022) when we try to mitigate exploratory data assumptions (e.g., all-policy concentrability). However, this framework suffers from a conundrum if only making weak realizability assumptions: we will know that the learned policy performs well only if we are informed with trajectories induced by it—rather than the ones induced by the covered policy.

More specifically, as the optimality of $\hat{\pi}$ could be quantified as $J^\star - J_{\hat{\pi}}$, the performance gap, we can telescope it through the performance difference lemma.

**Lemma 1** (The performance difference lemma). *We can decompose the performance gap as*

$$(1 - \gamma)(J_{\pi_1} - J_{\pi_2}) = \langle \mu_{\pi_1}, Q_{\pi_2}(\cdot, \pi_1) - Q_{\pi_2}(\cdot, \pi_2) \rangle.$$

Thus, with Eq. (4), if we want $J^\star - J_{\hat{\pi}}$ (i.e., $J_{\pi_e^\star} - J_{\hat{\pi}}$) to be equal to zero, it might be necessary to require $\mu_c$ to cover $\mu_{\hat{\pi}}$ ($\mu_c \gg \mu_{\hat{\pi}}$) since the right part of the inner product in Eq. (4) is always non-positive. However, as $\hat{\pi}$ is estimated and is even random when considering approximating it from a dataset, $\mu_c \gg \mu_{\hat{\pi}}$ is usually achieved through *all-policy concentrability*—$\mu_c \gg \mu_\pi$ for all $\pi$ in the hypothesis class. Single optimal concentrability fails to provide the desired result.

For instance, consider the counterexample in Figure 1 which is adapted from Zhan et al. (2022); Chen & Jiang (2022). Suppose we finally get the following covering distribution and policy:

$$\mu_c(s) = \begin{cases} 1/2 & \text{if } s = 1 \\ 1/2 & \text{if } s = 2 \end{cases} \quad \text{and} \quad \hat{\pi}(s) = \begin{cases} \text{L} & \text{if } s = 1 \\ \text{R} & \text{if } s = 3 \\ \text{Random} & \text{elsewhere.} \end{cases}$$

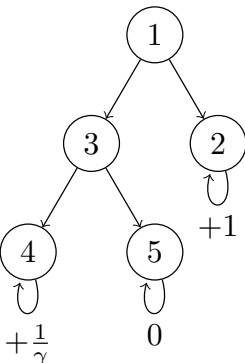

Figure 1: The above MDP is deterministic, and we initially start from state 1. We can take actions $L$ (left) and $R$ (right) in each state. In states 1 and 3, action $L$ ($R$) will transfer us to its left (right) hand state, and taking actions in other states will only cause a self-loop. We can only obtain non-zero rewards by taking actions in states 2 and 4, with values 1 and $\frac{1}{\gamma}$ correspondingly. There are two trajectories that could lead to the optimal $\gamma/(1-\gamma)$ return: $\{(1, \mathrm{R}), 2, \dots\}$ and $\{(1, \mathrm{L}), (3, \mathrm{L}), 4 \dots\}$. We take $\gamma$ as the discount factor.

While $\mu_c$ achieves single optimal concentrability and $\hat{\pi}$ achieves the maximized value of $Q^\star$ in each state on the support of $\mu_c$, $\hat{\pi}$ is not an optimal policy since it would end up with 0 return.

**Proposition 2.** *Assuming that there exist $\hat{q}$, $\hat{\pi}$ and a state-action distribution $d$, such that (i; dataset coverage) $d \gg d_{\pi^\star}$ for an optimal policy $\pi^\star$; (ii; performance of the estimation oracle) $\|Q^\star - \hat{q}\|_d^\infty = 0$;[2] (iii; export $\hat{\pi}$ from $\hat{q}$) $\langle \hat{q}(\cdot, \pi_e^\star) - \hat{q}(\cdot, \hat{\pi}), \mu \rangle = 0$ (cf. Eq. 4; $\mu$ is the margin of d). There exist MDPs with above properties satisfied such that $J_{\pi^\star} - J_{\hat{\pi}} = 1/(1 - \gamma)$.*

**How gap assumptions avoid this** While both Chen & Jiang (2022) and Uehara et al. (2023) consider single optimal concentrability and weak realizability assumptions (Uehara et al. (2023) also assumes additional structures of the dataset), the gap (margin) assumptions ensure that only taking $\pi^\star$ as $\hat{\pi}$ could achieve Eq. (4). Chen & Jiang (2022) also shows that with the gap assumption, we can even use a value-based algorithm to derive a near-optimal policy without accurately estimating $Q^\star$. Nevertheless, these gap (margin) assumptions only apply for part of MDPs.

**Remark 1.** While the gap assumption made in Ozdaglar et al. (2022) is applicable to all MDPs, their sample complexity depends on a concentration coefficient appearing nowhere in their assumptions. Further, the upper bound of this constant is not even clarified.

## 3.2 All-optimal concentrability

While single optimal concentrability suffers the hardness revealed before, there is still an alternative for the exploratory covering $\mu_c$, which is shown in the following lemma:

**Lemma 3** (From advantage to optimality). *If $\mu_c$ covers all distributions induced by* non-stationary *optimal policies (i.e., $\mu_c \gg \mu_{\pi_{\mathrm{non}}^\star}$ for any $\pi_{\mathrm{non}}^\star$) and Eq. (4) holds, then $\hat{\pi}$ is optimal and $J_{\hat{\pi}} = J^\star$.*

**Remark 2.** Non-stationary policies are frequently employed in the analysis of offline RL (Munos, 2003; 2005; Scherrer & Lesner, 2012; Chen & Jiang, 2019; Liu et al., 2020). If we make the gap assumption, the "all non-stationary" requirement is discardable since the action in each state that could lead to the optimal return is unique.

**Remark 3.** Wang et al. (2022) also utilizes the all-optimal concentrability assumption, but they consider the tabular setting and they require additionally gap assumptions to achieve the near-optimal guarantees.

We now provide a short proof of Lemma 3, showing by induction that $\hat{\pi}_i$—the non-stationary policy that adopts $\hat{\pi}$ at the beginning 0-th to $i$-th (include the $i$-th) steps and then follows $\pi_e^\star$—is optimal.

---

[2]this is stronger than the $L^1$ norm

*Proof.* We first rewrite the telescoping equation in the performance difference lemma as

$$(1-\gamma)(J_{\hat{\pi}_i} - J^\star) = \langle \mu_{\hat{\pi}_i}, Q^\star(\cdot, \hat{\pi}_i) - Q^\star(\cdot, \pi_e^\star) \rangle \tag{5}$$

$$= \langle \mu_{\hat{\pi}_i}^{0:i}, Q^\star(\cdot, \hat{\pi}) - Q^\star(\cdot, \pi_e^\star) \rangle + \langle \mu_{\hat{\pi}_i}^{i+1:\infty}, Q^\star(\cdot, \pi_e^\star) - Q^\star(\cdot, \pi_e^\star) \rangle \tag{6}$$

$$= \langle \mu_{\hat{\pi}_i}^{0:i}, Q^\star(\cdot, \hat{\pi}) - Q^\star(\cdot, \pi_e^\star) \rangle \tag{7}$$

where $\mu_\pi^{i:j}$ denotes the $i$-th to $j$-th steps (include the $i$-th and $j$-th) part of $\mu_\pi$. Thus, the optimality of $\hat{\pi}_i$ only depends on the first 0-th to $i$-th steps, and $\hat{\pi}_i$ is optimal if this part is on the support of $\mu_c$. Now we inductively show that, for any natural number $i$, the initial 0-th to $i$-th steps part is covered:

- The step-0 part of $\mu_{\hat{\pi}}$ (i.e., $(1-\gamma)\mu_0$) is on the support of $\mu_c$ since there is some (non-stationary) optimal policy $\pi^\star$ covered by it,

$$\mu_c \gg \mu_{\pi^\star} \gg \mu_0.$$

  Therefore, $\langle \mu_{\hat{\pi}_0}^{0:0}, Q^\star(\cdot, \hat{\pi}) - Q^\star(\cdot, \pi_e^\star) \rangle = 0$. From Eq. (7), $\hat{\pi}_0$ is optimal.

- We next show that if $\hat{\pi}_i$ is optimal (which means that it's covered $\mu_c$), then the first 0-th to $(i+1)$-th steps part of $\mu_{\hat{\pi}}$ is covered by $\mu_c$, which means that $\hat{\pi}_{i+1}$ is optimal. This comes from the fact that the initial 0-th to $(i+1)$-th steps part of the state distribution induced by a policy only depends on its previous 0-th to $i$-th decisions:

$$\mu_c \gg \mu_{\hat{\pi}_i} \gg \mu_{\hat{\pi}_i}^{0:i+1} = \mu_{\hat{\pi}}^{0:i+1}.$$

  From Eq. (7), $\hat{\pi}_{i+1}$ is optimal.

Thus, for any $\epsilon > 0$, there exists natural number $i \geq \log_\gamma \frac{\epsilon}{V_{\max}}$ such that

$$J^\star - J_{\hat{\pi}} \leq J^\star - J_{\hat{\pi}}^{0:i} \leq J^\star - (J_{\hat{\pi}_i} - \gamma^{i+1}V_{\max}) \leq \gamma^{i+1}V_{\max} \leq \epsilon,$$

where $J_\pi^{i:j}$ denotes the $i$-th to $j$-th steps part of the return. Therefore, $\hat{\pi}$ is optimal. □

Consequently, instead of the exploratory data assumption, all non-stationary optimal coverage is sufficient for policy optimization.

### 3.3 Dealing with statistical error

While Lemma 3 is adequate at the population level (i.e., with an infinite amount of data), covering all non-stationary optimal policies is not enough when considering the empirical setting (i.e., with finite samples) due to the introduced statistical error. This motivates us to adapt Lemma 3 with a more refined $\mu_c$.

**Assumption 1** (All near-optimal concentrability). We are given with a covering distribution $d_c$ such that its state distribution part $\mu_c$ covers the distributions induced by any non-stationary $\varepsilon_c$ near-optimal policy $\tilde{\pi}$:

$$\left\| \frac{\mu_{\tilde{\pi}}}{\mu_c} \right\|_\infty \leq C_c, \quad \forall \tilde{\pi} \in \Pi_{\varepsilon_c, \text{non}}^\star. \tag{8}$$

We call a policy $\pi$ is $\varepsilon$ near-optimal if $J^\star - J_\pi \leq \varepsilon$ and denote $\Pi_{\varepsilon, \text{non}}^\star$ as the class of all non-stationary $\varepsilon$ near-optimal policies. We also define $\frac{0}{0} = 1$ to suppress the extreme cases. With this refined $\mu_c$, we can now derive the optimality of $\hat{\pi}$ even with some statistical errors.

**Lemma 4** (From advantage to optimality, with statistical errors). *If $\langle \mu_c, Q^\star(\cdot, \hat{\pi}) - Q^\star(\cdot, \pi^\star) \rangle \geq -\varepsilon$ , and Assumption 1 holds with $\varepsilon_c \geq \frac{C_c \varepsilon}{1-\gamma}$, $\hat{\pi}$ is $\frac{C_c \varepsilon}{1-\gamma}$ near-optimal.*

We defer the proof of this lemma to Appendix C.1.

**Remark 4** (The asymptotic property of $\varepsilon_c$). One of the most important properties of all near-optimal concentrability is that $\varepsilon_c$ depends on the statistical error, which decreases as the amount of data increases.

# 4 Algorithm and analysis

After discussing the minimum requirement of estimating $Q^\star$, this section will demonstrate how to fulfill it and accomplish the policy learning task. Our algorithm, which is based on the optimal value estimator from Uehara et al. (2023), follows the pseudocode in Algorithm 1.

---

**Algorithm 1:** VOPR (Value-Based Offline RL with Policy Ratio)

---

**Input** : Dataset $\mathcal{D}$, value function class $\mathcal{Q}$, distribution density ratio class $\mathcal{W}$, policy ratio function class $\mathcal{B}$, and covering distribution $d_c$

**1** Estimate the optimal value function $\hat{q}$ as

$$\hat{q} = \underset{q \in \mathcal{Q}}{\operatorname{argmin}} \max_{w \in \mathcal{W}} \hat{\mathcal{L}}(d_c, q, w) \tag{9}$$

where

$$\hat{\mathcal{L}}(d, q, w) := 0.5 \mathbb{E}_d[q^2(s,a)] + \frac{1}{N_{\mathcal{D}}} \sum_{(s,a,r,s') \in \mathcal{D}} \left[ w(s,a) \big[ \gamma \max q(s', \cdot) + r - q(s,a) \big] \right] \tag{10}$$

**2** Derive the approximated optimal policy ratio:

$$\hat{\beta} = \underset{\beta \in \mathcal{B}}{\operatorname{argmax}} \, \mathbb{E}_{\mu_c}[\hat{q}(s, \pi_\beta)]$$

**Output:** $\hat{\pi} = \pi_{\hat{\beta}}$

---

We organized the rest of this section as follows: we first discuss the trade-off provided by the additional covering distribution $d_c$ and how to deduce $d_c$ in reality in Section 4.1; we then provide the finite-sample analysis of Algorithm 1 and its proof sketch in Section 4.2; we finally conclude this section by comparing our algorithms with the others in Section 4.3.

We defer the main proofs in this section to Appendix D.

## 4.1 Data assumptions and trade-off

As investigated in recent works (Huang & Jiang, 2022; Uehara et al., 2023), value function estimation under a given distribution requires a dataset that contains trajectories rolled out from it. Thus, our data assumption is as follows.

**Assumption 2** (Partial concentrability from $d_c$)**.** The shift from $d^{\mathcal{D}}$ to the induced state-action distribution by $\pi_e^\star$ from $d_c$ is bounded:

$$\left\| \frac{d_{d_c, \pi_e^\star}}{d^{\mathcal{D}}} \right\|_\infty \le C_{\mathcal{D}}. \tag{11}$$

It is clear that with Assumption 2, $d_c$ is also covered by $d^{\mathcal{D}}$.

**Proposition 5.** *If Assumption 2 holds, by definition of $d_{d_c, \pi_e^\star}$,*

$$\left\| \frac{d_c}{d^{\mathcal{D}}} \right\|_\infty \le \left\| \frac{d_{d_c, \pi_e^\star}/(1-\gamma)}{d^{\mathcal{D}}} \right\|_\infty \le \frac{C_{\mathcal{D}}}{1-\gamma}.$$

We now clarify the order of the learning process: we are first given with a dataset $\mathcal{D}$ with some good properties; then we try to find a $d_c$ from the support of the state-action distribution of $\mathcal{D}$ through some inductive bias (with necessary approximation); finally, we apply Algorithm 1 with $\mathcal{D}$ and $d_c$.

The choice of $d_c$ constructs a trade-off between the knowledge about optimal policy and the requirement of data coverage. On the one hand, the most casual choice of $d_c$ is $d^{\mathcal{D}}$ (as in Uehara et al. (2023)), which

means we have no prior knowledge about optimal policies. Employing $d^{\mathcal{D}}$ as $d_c$ will not only requires the dataset to cover unnecessary suboptimal trajectories, but also makes the dataset non-monotonic (adding new data points to it would break this assumption). On the other hand, if we have perfect knowledge about optimal policies, Assumption 2 could be significantly alleviated. More concretely, if $d_c$ strictly consists of the state-action distribution of trajectories induced by near-optimal policies, our data assumption reduces to the per-step version of near-optimal concentrability.

**Lemma 6.** *If $d_c$ is a linear combination of the state-action distributions induced by non-stationary $\varepsilon$ near-optimal policies $\Pi^{\star}_{\varepsilon,\mathrm{non}}$ under a fixed probability measure $\lambda$:*

$$d_c = \int_{\Pi^{\star}_{\varepsilon,\mathrm{non}}} d_{\tilde{\pi}} d\lambda(\tilde{\pi}). \tag{12}$$

*And $d^{\mathcal{D}}$ covers all admissible distributions of $\Pi^{\star}_{\varepsilon,\mathrm{non}}$:*

$$\forall\ \tilde{\pi} \in \Pi^{\star}_{\varepsilon,\mathrm{non}},\ i \in \mathbb{N},\ \left\| \frac{d_{\tilde{\pi},i}}{d^{\mathcal{D}}} \right\|_{\infty} \leq C,$$

*where $d_{\pi,i}$ denotes the normalized distribution of the $i$-th step part of $d_{\pi}$. The distribution shift from $d^{\mathcal{D}}$ is bounded as*

$$\left\| \frac{d_{d_c,\pi^{\star}_e}}{d^{\mathcal{D}}} \right\|_{\infty} \leq C.$$

While the above case is impractical in reality, it reveals the power of this inductive bias: the more auxiliary information we obtain about optimal paths, the weaker coverage assumptions of the dataset are required.

### 4.2 Finite-sample guarantee

We now give the finite-sample guarantee of Algorithm 1, but before proceeding, we should state necessary function class assumptions. The first are the weak realizability assumptions:

**Assumption 3** (Realizability of $\mathcal{W}$). There exists state-action distribution density ratio $w^{\star} \in \mathcal{W}$ such that $w^{\star} \circ d^{\mathcal{D}} = (I - \gamma P_{\pi^{\star}_e})^{-1} d_c Q^{\star}$.

**Assumption 4** (Realizability of $\mathcal{B}$). There exists policy ratio $\beta^{\star} \in \mathcal{B}$ such that $\beta^{\star} \circ \pi_c = \pi^{\star}_e$ and for all $s \in \mathcal{S}, \int_{\mathcal{A}} \beta(s,a)\pi_c(s,a)d\nu(a) = 1$.

**Assumption 5** (Realizability of $\mathcal{Q}$). $\mathcal{Q}$ contains the optimal value function: $Q^{\star} \in \mathcal{Q}$.

On the other hand, we gather all the bounding assumptions here.

**Assumption 6** (Boundness of $\mathcal{Q}$). For any $q \in \mathcal{Q}$, we assume $q \in (\mathcal{S} \times \mathcal{A} \to [0, V_{\max}])$.

**Assumption 7** (Boundness of $\mathcal{B}$). For any $\beta \in \mathcal{B}$, we assume $\beta \in (\mathcal{S} \times \mathcal{A} \to [0, U_{\mathcal{B}}])$.

**Assumption 8** (Boundness of $\mathcal{W}$). For any $w \in \mathcal{W}$, we assume $w \in (\mathcal{S} \times \mathcal{A} \to [0, U_{\mathcal{W}}])$.

**Remark 5** (Validity). The invertibility of $I - \gamma P_{\pi^{\star}_e}$ is shown by Lemma 11 in Appendix B.1. While Assumptions 3 and 8 actually subsumes Assumption 2, we make it explicit for clarity of explanation. Assumption 4 implicitly assumes that $\pi_c$ covers $\pi^{\star}_e$, this can easily be done by directly choosing $\pi_b$ as $\pi_c$.

**Remark 6.** Although we include the normalization step in Assumption 4, this can also be achieved with some preprocessing steps.

**Remark 7.** There is an overlap in the above assumptions: we can derive a policy ratio class $\mathcal{B}$ directly from $\mathcal{W}$ and $\mathcal{Q}$.

With these prerequisites in place, we can finally state our finite-sample guarantee.

**Theorem 1** (Sample complexity of learning a near-optimal policy)**.** *If Assumptions 1, 2, 3, 4, 5, 6, 7 and 8 hold with $\varepsilon_c \geq \frac{4C_c U_{\mathcal{B}} \sqrt{\varepsilon_{\mathrm{stat}}}}{1-\gamma}$ where*

$$\varepsilon_{\mathrm{stat}} = U_{\mathcal{W}} V_{\max} \sqrt{\frac{2\log(2|\mathcal{Q}||\mathcal{W}|/\delta)}{N_{\mathcal{D}}}},$$

*then with probability at least $1-\delta$, the output $\hat{\pi}$ from Algorithm 1 is near-optimal:*

$$J^{\star} - J_{\hat{\pi}} \leq \frac{4C_c U_{\mathcal{B}} \sqrt{\varepsilon_{\mathrm{stat}}}}{1-\gamma}.$$

**Proof sketch of Theorem 1**    As we can obtain the near-optimality guarantee via Lemma 4, the remaining task is to approximate Eq. (4). This comes from the following two lemmas.

**Lemma 7** ($L^2$ error of $\hat{q}$ under $d_c$, adapted from theorem 2 in Uehara et al. (2023))**.** *If Assumptions 2, 3, 5, 6 and 8 hold, with probability at least $1-\delta$, the estimated $\hat{q}$ from Algorithm 1 satisfies*

$$\|\hat{q} - Q^{\star}\|_{d_c,2} \leq 2\sqrt{\varepsilon_{\mathrm{stat}}}.$$

**Lemma 8** (From $L^1$ distance to Eq. (4))**.** *If Assumptions 4 and 7 hold,*

$$\langle Q^{\star}(\cdot, \pi_e^{\star}) - Q^{\star}(\cdot, \hat{\pi}), \mu_c \rangle \leq 2U_{\mathcal{B}} \|\hat{q} - Q^{\star}\|_{d_c,1}.$$

Combine them, we have that with probability at least $1-\delta$,

$$\langle Q^{\star}(\cdot, \pi_e^{\star}) - Q^{\star}(\cdot, \hat{\pi}), \mu_c \rangle \leq 2U_{\mathcal{B}} \|\hat{q} - Q^{\star}\|_{d_c,1} \leq 2U_{\mathcal{B}} \|\hat{q} - Q^{\star}\|_{d_c,2} \leq 4U_{\mathcal{B}} \sqrt{\varepsilon_{\mathrm{stat}}}.$$

### 4.3 Comparison with related works

We now provide a brief comparison of our method with some related algorithms.

**Algorithms with gap assumptions**    Chen & Jiang (2022) and Uehara et al. (2023) assume that there are (soft) gaps in the optimal value function, which is only satisfied by part of MDPs, whereas our goal is to deal with general problems. Moreover, while our algorithm is based on the optimal value estimator proposed by Uehara et al. (2023), we use the policy ratio to ensure a finite distribution shift and our near-optimality guarantee does not require the soft margin assumption. Besides, Uehara et al. (2023) use $d^{\mathcal{D}}$ as $d_c$, assuming that the dataset covers the optimal trajectories from itself. This assumption is non-monotonic and hard to be satisfied in reality. Instead, we propose using an additional covering distribution $d_c$ as an alternative, which can effectively utilize the prior knowledge about the optimal trajectories and trade off the dataset requirement.

**Algorithms with behavior regularization**    Zhan et al. (2022) use behavior regularization to ensure that the learned policy is close to the dataset. Nevertheless, the regularization makes the optimality of the learned policy unprovable without careful control on the regularization. The SMQP from Uehara et al. (2023) also uses regularizations, but its sample complexity is polynomial w.r.t. $|\mathcal{A}|$.

**Algorithms with pessimism in the face of uncertainty**    These algorithms (e.g., Jiang & Huang (2020); Liu et al. (2020); Xie et al. (2021); Cheng et al. (2022); Zhu et al. (2023)) are often closely related to approximate dynamic programming (ADP). They "pessimistically" estimate the given policies and update (or choose) policies "pessimistically" with the estimated value functions. However, the evaluation step used in these algorithms always requires the strong realization of all candidate policies' value functions, which our algorithm avoids.

**Limitations of our algorithm**    On the one hand, the additional covering distribution may be hard to access in some scenarios, leading back to using $d^{\mathcal{D}}$ as $d_c$. On the other hand, although mitigated with increasing dataset size, the assumption of covering all near-optimal policies is still stronger than the classic single-optimal concentrability. In addition, the "non-stationary" coverage requirement is also somewhat restrictive.

## 5 Conclusion and further work

This paper present VOPR, a new MIS-based algorithm for offline RL with function approximations. VOPR is inspired by the optimal value estimator proposed in Uehara et al. (2023), and it circumvents the soft margin assumption in the original paper with the near-optimal coverage assumption. While it still works if using the data distribution as the covering distribution, VOPR can trade off data assumptions with more refined choices. Compared with other algorithms considering partial coverage, VOPR does not make strong function class assumptions and works under general MDPs. Finally, despite the successes, a refined additional covering distribution may be difficult to obtain, and the near-optimal coverage assumption is still stronger than single optimal concentrability. We leave them for further investigation.

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

# A    Notations

Table 2: Notations

| | |
|---|---|
| $\mathcal{S}$ | state space |
| $\mathcal{A}$ | action space |
| $\mathcal{Q}$ | state-action value function class |
| $\mathcal{W}$ | state-action distribution ratio function class |
| $\mathcal{B}$ | policy ratio function class |
| $\beta$ | members of $\mathcal{B}$ |
| $J_\pi$ | expected return of the policy $\pi$ from the initial state distribution $\mu_0$ |
| $V_\pi$ | state value function for policy $\pi$ |
| $Q_\pi$ | state-action value function for policy $\pi$ |
| $V^\star$ | optimal state value function |
| $Q^\star$ | optimal state-action value function |
| $\nu$ | uniform measure of $\mathcal{A}$, $\mathcal{S}$, or $\mathcal{S} \times \mathcal{A}$, depending on the context |
| $\mathcal{D}$ | dataset used in the algorithm |
| $d^{\mathcal{D}}$ | state-action distribution of dataset |
| $\mu^{\mathcal{D}}$ | state distribution of dataset |
| $\pi_b$ | behaviour policy |
| $d_c$ | the additional covering distribution |
| $\mu_c$ | state distribution (margin) of the additional covering distribution |
| $\pi_c$ | policy of the additional covering distribution |
| $\langle a, b \rangle$ | inner product of $a$ and $b$, usually as $\int ab\, d\nu$ |
| $f_1 \circ f_2$ | $(f_1 \circ f_2)(s,a) = f_1(s,a)f_2(s,a)$, normalizing it if needed (e.g., density) |
| $\mu \times \pi$ | $(\mu \times \pi)(s,a) = \mu(s)\pi(a \mid s)$ |
| $T^\star$ | Bellman optimality operator, $T^\star q(s,a) \coloneqq R(s,a) + \gamma \mathbb{E}_{s' \sim P(\cdot\mid s,a)}[\max q(s', \cdot)]$ |
| $\mu_0$ | initial state distribution |
| $\mu_\pi^{i:j}$ | the $i$-th to $j$-th steps part of $\mu_\pi$ |
| $d_1 \gg d_2$ | $d_2$ is absolutely continuous w.r.t. $d_1$ |
| $d_{\pi,i}$ | normalize $i$-th step part of state-action distribution induced by $\pi$ |
| $d_{d,\pi}$ | state-action distribution induced by $\pi$ from $d$ |
| $\pi_i$ | policy take $\pi$ in the previous 0-th to $i$-th (include the $i$-th) steps, and take $\pi_e^\star$ after this |
| $\pi_\beta$ | $\pi_\beta(a \mid s) = \pi_c(a \mid s)\beta(s,a) / \int_{\mathcal{A}} \pi_c(a \mid s)\beta(s,a)d\nu(a)$ |
| $\Pi_{\varepsilon,\mathrm{non}}^\star$ | the class of all non-stationary $\varepsilon$ near-optimal policies |
| $P_\pi$ | state-action transition kernel with policy $\pi$ |
| $O^\star$ | conjucate operator of some operator $O$ |

While $\pi, \mu$, and $d$ are mainly used to denote the Radon–Nikodym derivatives of the underlying probability measures w.r.t. $\nu$, we sometimes also use them to represent the correspondent distribution measure with abuse of notation.

# B    Helper Lemmas

## B.1    Properties of $P_\pi$

We first provide some properties of $P_\pi$ (for any policy $\pi$) as an operator on the $L^\infty$-space of $\mathcal{S} \times \mathcal{A}$, and similar results should also hold for transition operators with policies defined on $\mathcal{S}$. Note that the integrations of the absolute value of the functions considered in this subsection are always finite, which means that we can change the orders of integrations via Fubini's Theorem. As we will consider conjugate operators, we define the inner product as $\langle q, d \rangle = \int_{\mathcal{S} \times \mathcal{A}} q(s,a)d(s,a)d\nu(s,a)$.

**Lemma 9.** $P_\pi$ *is* linear.

*Proof.* Recall the definition of $P_\pi$,

$$P_\pi d(s', a') = \int_{\mathcal{S} \times \mathcal{A}} \pi(a' \mid s') P(s' \mid s, a) d(s, a) d\nu(s, a)$$

For any $d_1, d_2 \in L^\infty(\mathcal{S} \times \mathcal{A})$,

$$
\begin{aligned}
P_\pi(\alpha_1 d_1 + \alpha_2 d_2)(s', a') &= \int_{\mathcal{S} \times \mathcal{A}} \pi(a' \mid s') P(s' \mid s, a)(\alpha_1 d_1 + \alpha_2 d_2)(s, a) d\nu(s, a) \\
&= \int_{\mathcal{S} \times \mathcal{A}} \alpha_1 \pi(a' \mid s') P(s' \mid s, a) d_1(s, a) d\nu(s, a) + \int_{\mathcal{S} \times \mathcal{A}} \alpha_2 \pi(a' \mid s') P(s' \mid s, a) d_2(s, a) d\nu(s, a) \\
&= \alpha_1 \int_{\mathcal{S} \times \mathcal{A}} \pi(a' \mid s') P(s' \mid s, a) d_1(s, a) d\nu(s, a) + \alpha_2 \int_{\mathcal{S} \times \mathcal{A}} \pi(a' \mid s') P(s' \mid s, a) d_2(s, a) d\nu(s, a) \\
&= \alpha_1 P_\pi d_1(s', a') + \alpha_2 P_\pi d_2(s', a')
\end{aligned}
$$

This compeletes the proof. $\qquad\square$

**Lemma 10.** *The adjoint operator of $P_\pi$ is*

$$P_\pi^\star q(s, a) = \int_{\mathcal{S} \times \mathcal{A}} q(s', a') \pi(a' \mid s') P(s' \mid s, a) d\nu(s', a').$$

**Remark 8.** Intuitively, we can see $P_\pi d(s', a')$ as one-step forward of $d$, such that we start from $(s, a) \sim d$, transit into $s' \sim P(\cdot \mid s, a)$ and take $a' \sim \pi(\cdot \mid s')$. Also, we can view $P_\pi^\star q(s, a)$ as one-step backward of $q$, such that we compute the value of $(s, a)$ through the one step transferred state-action distribution with the help of $q$.

*Proof.* Consider the inner products $\langle q, P_\pi d \rangle$ and $\langle P_\pi^\star q, d \rangle$, we should prove that these two are equal. By definition,

$$
\begin{aligned}
\langle q, P_\pi d \rangle &= \int_{\mathcal{S} \times \mathcal{A}} \left[ q(s', a') \int_{\mathcal{S} \times \mathcal{A}} \pi(a' \mid s') P(s' \mid s, a) d(s, a) d\nu(s, a) \right] d\nu(s', a') \\
&= \int_{\mathcal{S} \times \mathcal{A}} \int_{\mathcal{S} \times \mathcal{A}} q(s', a') \pi(a' \mid s') P(s' \mid s, a) d(s, a) d\nu(s, a) d\nu(s', a')
\end{aligned}
$$

and

$$
\begin{aligned}
\langle P_\pi^\star q, d \rangle &= \int_{\mathcal{S} \times \mathcal{A}} d(s, a) \left[ \int_{\mathcal{S} \times \mathcal{A}} q(s', a') \pi(a' \mid s') P(s' \mid s, a) d\nu(s', a') \right] d\nu(s, a) \\
&= \int_{\mathcal{S} \times \mathcal{A}} \int_{\mathcal{S} \times \mathcal{A}} d(s, a) q(s', a') \pi(a' \mid s') P(s' \mid s, a) d\nu(s', a') d\nu(s, a) \\
&= \int_{\mathcal{S} \times \mathcal{A}} \int_{\mathcal{S} \times \mathcal{A}} q(s', a') \pi(a' \mid s') P(s' \mid s, a) d(s, a) d\nu(s, a) d\nu(s', a'). \qquad \text{(Fubini's Theorem)}
\end{aligned}
$$

This completes the proof. $\qquad\square$

**Lemma.** $\|P_\pi^\star\|_\infty = \|P_\pi\|_\infty \leq 1$

**Remark 9.** This upper bound should be intuitive since that $P_\pi$ can be seen as a probability transition kernel from $\mathcal{S} \times \mathcal{A}$ to itself.

*Proof.* Fix any $s \in \mathcal{S}$, $a \in \mathcal{A}$, we define $p(s', a') = P(s' \mid s, a)\pi(a' \mid s')$, By Fubini's theorem, we have that

$$
\begin{aligned}
\|p\|_{1,\nu} = \int_{\mathcal{S} \times \mathcal{A}} |p| d\nu &= \int_{\mathcal{S} \times \mathcal{A}} p d\nu \\
&= \int_{\mathcal{S}} \int_{\mathcal{A}} P(s' \mid s, a)\pi(a' \mid s') d\nu(a') d\nu(s') \\
&= \int_{\mathcal{S}} P(s' \mid s, a) \left[ \int_{\mathcal{A}} \pi(a' \mid s') d\nu(a') \right] d\nu(s') \\
&= \int_{\mathcal{S}} P(s' \mid s, a) d\nu(s') \\
&= 1.
\end{aligned}
$$

For another function $q$ on $\mathcal{S} \times \mathcal{A}$ such that $\|q\|_{\infty,\nu} \leq 1$, we can use Hölder's inequality, which yields

$$
\|pq\|_{1,\nu} \leq \|q\|_{\infty,\nu} \|p\|_{1,\nu} \leq 1.
$$

Thus, for any $s \in \mathcal{S}, a \in \mathcal{A}$, and function $q$ with $\|q\|_{\infty,\nu} \leq 1$,

$$
P_\pi^\star q(s, a) = \int_{\mathcal{S} \times \mathcal{A}} q(s', a')\pi(a' \mid s')P(s' \mid s, a) d\nu(s', a') = \|pq\|_{1,\nu} \leq 1.
$$

So, we have that

$$
\|P_\pi\|_\infty = \|P_\pi^\star\|_\infty = \max_{\|q\|_\infty \leq 1} \|P_\pi^\star q\|_{\infty,\nu} \leq \max_{\|q\|_\infty \leq 1} \max_{s \in \mathcal{S}, a \in \mathcal{A}} P_\pi^\star q(s, a) \leq 1.
$$

This completes the proof. $\qquad\square$

**Lemma 11.** *$I - \gamma P_\pi$ is invertible and*

$$
(I - \gamma P_\pi)^{-1} = \sum_{i=0}^{\infty} (\gamma P_\pi)^i.
$$

*Proof.* Since $\|P_\pi\|_\infty \leq 1$, $\sum_{i=0}^{\infty} (\gamma P_\pi)^i$ converges. Take multiplication

$$
\begin{aligned}
(I - \gamma P_\pi)[\sum_{i=0}^{\infty} (\gamma P_\pi)^i] &= \sum_{i=0}^{\infty} (\gamma P_\pi)^i - \sum_{i=1}^{\infty} (\gamma P_\pi)^i \\
&= (\gamma P_\pi)^0 \\
&= I.
\end{aligned}
$$

This completes the proof. $\qquad\square$

**Proposition 12.** *By definition, $d_{d,\pi} = (1 - \gamma) \sum_{i=0}^{\infty} (\gamma P_\pi)^i d = (1 - \gamma)(I - \gamma P_\pi)^{-1} d$.*

## B.2 Other useful lemmas

**Lemma** (Performance difference lemma)**.** *We can decompose the performance gap as*

$$
(1 - \gamma)(J_{\pi_1} - J_{\pi_2}) = \langle \mu_{\pi_1}, Q_{\pi_2}(\cdot, \pi_1) - Q_{\pi_2}(\cdot, \pi_2) \rangle.
$$

*Proof.* By definition,

$$
\begin{aligned}
\langle \mu_{\pi_1}, Q_{\pi_2}(\cdot, \pi_1) - Q_{\pi_2}(\cdot, \pi_2) \rangle =& \mathbb{E}_{s \sim \mu_{\pi_1}} \big[ R(\cdot, \pi_1) + \gamma \mathbb{E}_{a \sim \pi_1(\cdot|s), s' \sim P(\cdot|s,a)} [Q_{\pi_2}(s', \pi_2)] - Q_{\pi_2}(\cdot, \pi_2) \rangle \big] \\
=& \mathbb{E}_{s \sim \mu_{\pi_1}} \big[ R(\cdot, \pi_1) \big] + \mathbb{E}_{s \sim \mu_{\pi_1}} \big[ \gamma \mathbb{E}_{a \sim \pi_1(\cdot|s), s' \sim P(\cdot|s,a)} [Q_{\pi_2}(s', \pi_2)] \big] \\
& - \mathbb{E}_{s \sim \mu_{\pi_1}} \big[ Q_{\pi_2}(\cdot, \pi_2) \rangle \big] \\
=& \mathbb{E}_{s \sim \mu_{\pi_1}} \big[ R(\cdot, \pi_1) \big] + \gamma \mathbb{E}_{s \sim \mu_{\pi_1}, a \sim \pi_1(\cdot|s), s' \sim P(\cdot|s,a)} [Q_{\pi_2}(s', \pi_2)] \big] \\
& - \mathbb{E}_{s \sim \mu_{\pi_1}} \big[ Q_{\pi_2}(\cdot, \pi_2) \rangle \big] \\
=& \mathbb{E}_{s \sim \mu_{\pi_1}} \big[ R(\cdot, \pi_1) \big] + \mathbb{E}_{s \sim \mu_{\pi_1}} [Q_{\pi_2}(s, \pi_2)] \big] - (1-\gamma) \mathbb{E}_{s \sim \mu_0} [Q_{\pi_2}(s, \pi_2)] \big] \\
& - \mathbb{E}_{s \sim \mu_{\pi_1}} \big[ Q_{\pi_2}(\cdot, \pi_2) \rangle \big] \\
=& \mathbb{E}_{s \sim \mu_{\pi_1}} \big[ R(\cdot, \pi_1) \big] - (1-\gamma) \mathbb{E}_{s \sim \mu_0} [Q_{\pi_2}(s, \pi_2)] \big] \\
=& (1-\gamma)(J_{\pi_1} - J_{\pi_2})
\end{aligned}
$$

The first equality comes from Bellman equation, and the fourth equality comes from the definition of $\mu_\pi$. This completes the proof. □

## C  Detailed proofs for Section 3

### C.1  Proof of Lemma 4

**Lemma** (From advantage to optimality, restatement of Lemma 4)**.** *If $\langle \mu_c, Q^\star(\cdot, \hat{\pi}) - Q^\star(\cdot, \pi^\star) \rangle \geq -\varepsilon$ , and Assumption 1 holds with $\varepsilon_c \geq \frac{C_c \varepsilon}{1-\gamma}$, $\hat{\pi}$ is $\frac{C_c \varepsilon}{1-\gamma}$ near-optimal.*

*Proof.* We begin with using induction to prove that $\hat{\pi}_i$ is $\frac{C_c \varepsilon}{1-\gamma}$ near-optimal for any $i \in \mathbb{N}$:

- We first show that $\hat{\pi}_0$ is $\frac{C_c \varepsilon}{1-\gamma}$ near-optimal. From Assumption 1, we can use any $\tilde{\pi} \in \Pi^\star_{\varepsilon_c, \text{non}}$ to conclude that

$$
\left\| \frac{\mu_0}{\mu_c} \right\|_\infty \leq \left\| \frac{\mu_{\tilde{\pi}}/(1-\gamma)}{\mu_c} \right\|_\infty \leq \frac{C_c}{1-\gamma}.
$$

Thus, we can the show optimality of $\hat{\pi}_0^\star$ by the advantage:

$$
\begin{aligned}
\langle \mu_{\hat{\pi}_0}, Q^\star(\cdot, \hat{\pi}_0) - Q^\star(\cdot, \pi_e^\star) \rangle =& \langle \mu_{\hat{\pi}_0}^{0:0}, Q^\star(\cdot, \hat{\pi}) - Q^\star(\cdot, \pi_e^\star) \rangle + \langle \mu_{\hat{\pi}_0}^{1:\infty}, Q^\star(\cdot, \pi_e^\star) - Q^\star(\cdot, \pi_e^\star) \rangle \\
=& \langle \mu_{\hat{\pi}_0^\star}^{0:0}, Q^\star(\cdot, \hat{\pi}) - Q^\star(\cdot, \pi_e^\star) \rangle \\
=& (1-\gamma) \langle \mu_0, Q^\star(\cdot, \hat{\pi}) - Q^\star(\cdot, \pi_e^\star) \rangle \\
\geq& C_c \langle \mu_c, Q^\star(\cdot, \hat{\pi}) - Q^\star(\cdot, \pi_e^\star) \rangle \quad (Q^\star(\cdot, \hat{\pi}) - Q^\star(\cdot, \pi_e^\star) \text{ is non-positive}) \\
\geq& - C_c \varepsilon.
\end{aligned}
$$

By performance difference lemma,

$$
\begin{aligned}
(1-\gamma)(J_{\hat{\pi}_0} - J^\star) =& \langle \mu_{\hat{\pi}_0}, Q^\star(\cdot, \hat{\pi}_0) - Q^\star(\cdot, \pi_e^\star) \rangle \\
\geq& - C_c \varepsilon.
\end{aligned}
$$

- Next, we show that if $\hat{\pi}_i$ is $\frac{C_c \varepsilon}{1-\gamma}$ near-optimal, $\hat{\pi}_{i+1}$ is $\frac{C_c \varepsilon}{1-\gamma}$ near-optimal. Since that $\hat{\pi}_i$ is $\frac{C_c \varepsilon}{1-\gamma}$ optimal, the distribution shift from $\mu_c$ to $\mu_{\hat{\pi}_i}$ is bounded, which means,

$$
\left\| \frac{\mu_{\hat{\pi}}^{0:i+1}}{\mu_c} \right\|_\infty = \left\| \frac{\mu_{\hat{\pi}_i}^{0:i+1}}{\mu_c} \right\|_\infty \leq \left\| \frac{\mu_{\hat{\pi}_i}}{\mu_c} \right\|_\infty \leq C_c.
$$

Then, we have

$$
\begin{aligned}
&\langle \mu_{\hat{\pi}_{i+1}}, Q^\star(\cdot, \hat{\pi}_{i+1}) - Q^\star(\cdot, \pi_e^\star) \rangle \\
=&\langle \mu_{\hat{\pi}_{i+1}}^{0:i+1}, Q^\star(\cdot, \hat{\pi}) - Q^\star(\cdot, \pi_e^\star) \rangle + \langle \mu_{\hat{\pi}_{i+1}}^{i+2:\infty}, Q^\star(\cdot, \pi_e^\star) - Q^\star(\cdot, \pi_e^\star) \rangle \\
=&\langle \mu_{\hat{\pi}_{i+1}}^{0:i+1}, Q^\star(\cdot, \hat{\pi}) - Q^\star(\cdot, \pi_e^\star) \rangle \\
=&\langle \mu_{\hat{\pi}}^{0:i+1}, Q^\star(\cdot, \hat{\pi}) - Q^\star(\cdot, \pi_e^\star) \rangle \\
\geq& C_c \langle \mu_c, Q^\star(\cdot, \hat{\pi}) - Q^\star(\cdot, \pi_e^\star) \rangle \qquad\qquad (Q^\star(\cdot, \hat{\pi}) - Q^\star(\cdot, \pi_e^\star) \text{ is non-positive}) \\
\geq& -C_c \varepsilon.
\end{aligned}
$$

By performance difference lemma,

$$
\begin{aligned}
(1-\gamma)(J_{\hat{\pi}_{i+1}} - J^\star) =& \langle \mu_{\hat{\pi}_{i+1}}, Q^\star(\cdot, \hat{\pi}_{i+1}) - Q^\star(\cdot, \pi_e^\star) \rangle. \\
\geq& -C_c \varepsilon
\end{aligned}
$$

Therefore, $\hat{\pi}_{i+1}$ is $\frac{C_c \varepsilon}{1-\gamma}$ near-optimal.

Thus, for any $\epsilon > 0$, there exists natural number $i \geq \log_\gamma \frac{\epsilon}{V_{\max}}$ such that

$$
J^\star - J_{\hat{\pi}} \leq J^\star - J_{\hat{\pi}}^{0:i} \leq J^\star - (J_{\hat{\pi}_i} - \gamma^{i+1} V_{\max}) \leq \frac{C_c \varepsilon}{1-\gamma} + \gamma^{i+1} V_{\max} \leq \frac{C_c \varepsilon}{1-\gamma} + \epsilon,
$$

where $J_\pi^{i:j}$ denotes the $i$-th to $j$-th steps part of the return. Therefore, $\hat{\pi}$ is $\frac{C_c \varepsilon}{1-\gamma}$ near-optimal. $\square$

## D  Detailed proofs for Section 4

### D.1  Proof of Lemma 6

**Lemma** (Restatement of Lemma 6). *If $d_c$ is a linear combination of the state-action distributions induced by $\varepsilon$ near-optimal non-stationary policies $\Pi_{\varepsilon,\mathrm{non}}^\star$ under a fixed probability measure $\lambda$:*

$$
d_c = \int_{\Pi_{\varepsilon,\mathrm{non}}^\star} d_{\tilde{\pi}} d\lambda(\tilde{\pi}). \tag{13}
$$

*And $d^{\mathcal{D}}$ covers all admissible distributions of $\Pi_{\varepsilon,\mathrm{non}}^\star$:*

$$
\forall\, \tilde{\pi} \in \Pi_{\varepsilon,\mathrm{non}}^\star,\; i \in \mathbb{N},\; \left\| \frac{d_{\tilde{\pi},i}}{d^{\mathcal{D}}} \right\|_\infty \leq C.
$$

*The distribution shift from $d^{\mathcal{D}}$ is bounded as*

$$
\left\| \frac{d_{d_c, \pi_e^\star}}{d^{\mathcal{D}}} \right\|_\infty \leq C.
$$

*Proof.* Define the state-action distribution of policy $\pi$ from $s \in \mathcal{S}, a \in \mathcal{A}$ at step $i$ as

$$
\begin{aligned}
d_{s,a,\pi,i}(s', a') = P(s_i = s', a_i = a' \,| s_0 = s, a_0 = a, s_1 \sim P(\cdot \mid s_0, a_0), a_1 \sim \pi(\cdot \mid s_1) \dots \\
s_j \sim P(\cdot \mid s_{j-1}, a_{j-1}), a_j \sim \pi(\cdot \mid s_j) \dots).
\end{aligned}
$$

Also, define the global version of it as

$$
d_{s,a,\pi}(s', a') = (1-\gamma) \sum_{i=0}^{\infty} d_{s,a,\pi,i}(s', a').
$$

We can rewrite $d_{d_c, \pi_e^\star}(s, a)$ as

$$
\begin{aligned}
d_{d_c, \pi_e^\star}(s, a) &= \int_{\mathcal{S} \times \mathcal{A}} d_{s_1, a_1, \pi_e^\star}(s, a) d_c(s_1, a_1) d\nu(s_1, a_1) \\
&= \int_{\mathcal{S} \times \mathcal{A}} d_{s_1, a_1, \pi_e^\star}(s, a) \Big[ \int_\Pi d_{\tilde{\pi}}(s_1, a_1) d\lambda(\tilde{\pi}) \Big] d\nu(s_1, a_1) \\
&= \int_\Pi \Big[ \int_{\mathcal{S} \times \mathcal{A}} d_{s_1, a_1, \pi_e^\star}(s, a) d_{\tilde{\pi}}(s_1, a_1) d\nu(s_1, a_1) \Big] d\lambda(\tilde{\pi}) \qquad \text{(Fubini's Theorem)} \\
&= \int_\Pi \Big[ \int_{\mathcal{S} \times \mathcal{A}} (1 - \gamma) \sum_{i=0}^\infty \big[ \gamma^i d_{s_1, a_1, \pi_e^\star}(s, a) d_{\tilde{\pi}, i}(s_1, a_1) \big] d\nu(s_1, a_1) \Big] d\lambda(\tilde{\pi}) \\
&= \int_\Pi \Big[ (1 - \gamma) \sum_{i=0}^\infty \big[ \gamma^i \int_{\mathcal{S} \times \mathcal{A}} d_{s_1, a_1, \pi_e^\star}(s, a) d_{\tilde{\pi}, i}(s_1, a_1) d\nu(s_1, a_1) \big] \Big] d\lambda(\tilde{\pi}) \\
&= \int_\Pi \Big[ (1 - \gamma) \sum_{i=0}^\infty d_{\tilde{\pi}_i}^{i:\infty}(s, a) \Big] d\lambda(\tilde{\pi}).
\end{aligned}
$$

The last equation comes from that

$$
\begin{aligned}
&\gamma^i \int_{\mathcal{S} \times \mathcal{A}} d_{s_1, a_1, \pi_e^\star}(s, a) d_{\tilde{\pi}, i}(s_1, a_1) d\nu(s_1, a_1) \\
=& \gamma^i \int_{\mathcal{S} \times \mathcal{A}} d_{s_1, a_1, \pi_e^\star}(s, a) \Big[ \int_{\mathcal{S}} \Big[ \int_{\mathcal{A}} d_{s_2, a_2, \tilde{\pi}, i}(s_1, a_1) \tilde{\pi}(a_2 \mid s_2) d\nu(a_2) \Big] \mu_0(s_2) d\nu(s_2) \Big] d\nu(s_1, a_1) \\
=& \int_{\mathcal{S}} \Big[ \int_{\mathcal{A}} \Big[ \gamma^i \int_{\mathcal{S} \times \mathcal{A}} d_{s_1, a_1, \pi_e^\star}(s, a) d_{s_2, a_2, \tilde{\pi}, i}(s_1, a_1) d\nu(s_1, a_1) \Big] \tilde{\pi}(a_2 \mid s_2) d\nu(a_2) \Big] \mu_0(s_2) d\nu(s_2),
\end{aligned}
$$
$$
\text{(Fubini's Theorem)}
$$

since

$$
\begin{aligned}
&\gamma^i \int_{\mathcal{S} \times \mathcal{A}} d_{s_1, a_1, \pi_e^\star}(s, a) d_{s_2, a_2, \tilde{\pi}, i}(s_1, a_1) d\nu(s_1, a_1) \\
=& \gamma^i \int_{\mathcal{S} \times \mathcal{A}} (1 - \gamma) \sum_{k=0}^\infty \big[ \gamma^k d_{s_1, a_1, \pi_e^\star, k}(s, a) \big] d_{s_2, a_2, \tilde{\pi}, i}(s_1, a_1) d\nu(s_1, a_1) \\
=& (1 - \gamma) \sum_{k=0}^\infty \Big[ \gamma^{k+i} \int_{\mathcal{S} \times \mathcal{A}} d_{s_1, a_1, \pi_e^\star, k}(s, a) d_{s_2, a_2, \tilde{\pi}, i}(s_1, a_1) d\nu(s_1, a_1) \Big] \\
=& (1 - \gamma) \sum_{k=0}^\infty \big[ \gamma^{k+i} d_{s_2, a_2, \tilde{\pi}_i, k+i}(s, a) \big] \\
=& (1 - \gamma) \sum_{k=i}^\infty \big[ \gamma^k d_{s_2, a_2, \tilde{\pi}_i, k}(s, a) \big] \\
=& d_{s_2, a_2, \tilde{\pi}_i}^{i:\infty}(s, a),
\end{aligned}
$$

we get

$$
\begin{aligned}
&\gamma^i \int_{\mathcal{S} \times \mathcal{A}} d_{s_1, a_1, \pi_e^\star}(s, a) d_{\tilde{\pi}, i}(s_1, a_1) d\nu(s_1, a_1) \\
=& \int_{\mathcal{S}} \Big[ \int_{\mathcal{A}} \big[ d_{s_2, a_2, \tilde{\pi}_i}^{i:\infty}(s, a) \big] \tilde{\pi}(a_2 \mid s_2) d\nu(a_2) \Big] \mu_0(s_2) d\nu(s_2) \\
=& d_{\tilde{\pi}_i}^{i:\infty}(s, a).
\end{aligned}
$$

Finally, $\forall s \in \mathcal{S}, a \in \mathcal{A}$,

$$
\begin{aligned}
\frac{d_{d_c, \pi_e^\star}(s,a)}{d^{\mathcal{D}}(s,a)} &= \int_{\Pi} \Big[ (1-\gamma) \sum_{i=0}^{\infty} \frac{d_{\tilde{\pi}_i}^{i:\infty}(s,a)}{d^{\mathcal{D}}(s,a)} \Big] d\lambda(\tilde{\pi}) \\
&= \int_{\Pi} \Big[ (1-\gamma) \sum_{i=0}^{\infty} \frac{(1-\gamma) \sum_{j=i}^{\infty} \gamma^j d_{\tilde{\pi}_i, j}(s,a)}{d^{\mathcal{D}}(s,a)} \Big] d\lambda(\tilde{\pi}) \\
&= \int_{\Pi} \Big[ (1-\gamma) \sum_{i=0}^{\infty} (1-\gamma) \sum_{j=i}^{\infty} \gamma^j \frac{d_{\tilde{\pi}_i, j}(s,a)}{d^{\mathcal{D}}(s,a)} \Big] d\lambda(\tilde{\pi}) \\
&\leq \int_{\Pi} \Big[ C(1-\gamma)^2 \sum_{i=0}^{\infty} \sum_{j=i}^{\infty} \gamma^j \Big] d\lambda(\tilde{\pi}) \qquad (\tilde{\pi} \in \Pi_{\varepsilon,\mathrm{non}}^\star \text{ indicates } \tilde{\pi}_i \in \Pi_{\varepsilon,\mathrm{non}}^\star) \\
&\leq \int_{\Pi} \Big[ C(1-\gamma)^2 \sum_{i=0}^{\infty} \frac{\gamma^i}{1-\gamma} \Big] d\lambda(\tilde{\pi}) \\
&\leq \int_{\Pi} C d\lambda(\tilde{\pi}) \\
&= C.
\end{aligned}
$$

This completes the proof. $\qquad\qquad\qquad\qquad\qquad\qquad\qquad\qquad\qquad\qquad\qquad\qquad\qquad\quad$ $\square$

### D.2 Proof of Lemma 7

Note that the lemmas and proofs of this subsection are mainly adapted from Uehara et al. (2023), similar statements could also be found in the original paper. However, since that we use $d_c$ to replace $d^{\mathcal{D}}$, we present them for clarity of explanation and to make our paper self-contained. We refer interested readers to the original paper for another detail.

We first define the expected version of Eq. (10) as

$$
\begin{aligned}
\mathcal{L}(d, q, w) :=& 0.5 \mathbb{E}_d[q^2(s,a)] + \mathbb{E}_{(s,a) \sim d_w^{\mathcal{D}}, r=R(s,a), s' \sim P(\cdot|s,a)} \big[ \gamma \max q(s', \cdot) + r - q(s,a) \big] \\
=& 0.5 \mathbb{E}_d[q^2(s,a)] + \mathbb{E}_{\mathcal{D}_w} \big[ \gamma \max q(s', \cdot) + r - q(s,a) \big]
\end{aligned}
$$

where $d_w^{\mathcal{D}} = d^{\mathcal{D}} \circ w$, and $\mathbb{E}_{\mathcal{D}_w}$ denotes taking expectation with respect to the reweighted data collecting process.

**Lemma 13** (Expectation)**.** *The expected value of $\hat{\mathcal{L}}(d, q, w)$ w.r.t. the data collecting process is $\mathcal{L}(d, q, w)$:*

$$
\mathbb{E}_{\mathcal{D}}[\hat{\mathcal{L}}(d, q, w)] = \mathcal{L}(d, q, w).
$$

*Proof.* Since only the second term of $\hat{\mathcal{L}}$ is random, we additional define

$$
\hat{\mathcal{L}}_{\mathcal{W}}(q, w) := \frac{1}{N_{\mathcal{D}}} \sum_{(s,a,r,s') \in \mathcal{D}} \mathbb{E}_{\mathcal{D}} \Big[ w(s,a) \big[ \gamma \max q(s', \cdot) + r - q(s,a) \big].
$$

We can rearrange the expectation as follows,

$$
\begin{aligned}
\mathbb{E}_{\mathcal{D}}[\hat{\mathcal{L}}(d, q, w)] =& \mathbb{E}_{\mathcal{D}} \Big[ 0.5 \mathbb{E}_d[q^2(s,a)] + \hat{\mathcal{L}}_{\mathcal{W}}(q, w) \Big] & (14) \\
=& \mathbb{E}_{\mathcal{D}} \Big[ 0.5 \mathbb{E}_d[q^2(s,a)] \Big] + \mathbb{E}_{\mathcal{D}} \Big[ \hat{\mathcal{L}}_{\mathcal{W}}(q, w) \Big] & (15) \\
=& 0.5 \mathbb{E}_d[q^2(s,a)] + \mathbb{E}_{\mathcal{D}} \Big[ \hat{\mathcal{L}}_{\mathcal{W}}(q, w) \Big] & (16)
\end{aligned}
$$

Then, by the i.i.d. assumption of samples and linear property of MIS,

$$
\begin{aligned}
\mathbb{E}_{\mathcal{D}}[\hat{\mathcal{L}}(d,q,w)] =& 0.5\mathbb{E}_d[q^2(s,a)] + \mathbb{E}_{\mathcal{D}}\left[\frac{1}{N_{\mathcal{D}}}\sum_{(s,a,r,s')\in\mathcal{D}}\left[w(s,a)\left[\gamma\max q(s',\cdot)+r-q(s,a)\right]\right]\right] \\
=& 0.5\mathbb{E}_d[q^2(s,a)] + \frac{1}{N_{\mathcal{D}}}\sum_{(s,a,r,s')\in\mathcal{D}}\mathbb{E}_{\mathcal{D}}\left[w(s,a)\left[\gamma\max q(s',\cdot)+r-q(s,a)\right]\right] \\
=& 0.5\mathbb{E}_d[q^2(s,a)] + \mathbb{E}_{\mathcal{D}}\left[w(s,a)\left[\gamma\max q(s',\cdot)+r-q(s,a)\right]\right] \\
=& 0.5\mathbb{E}_d[q^2(s,a)] + \mathbb{E}_{(s,a)\sim d^{\mathcal{D}},r=R(s,a),s'\sim P(\cdot|s,a)}\left[w(s,a)\left[\gamma\max q(s',\cdot)+r-q(s,a)\right]\right] \\
=& 0.5\mathbb{E}_d[q^2(s,a)] + \mathbb{E}_{(s,a)\sim d^{\mathcal{D}}}\left[w(s,a)\left[\mathbb{E}_{s'\sim P(\cdot|s,a)}[\gamma\max q(s',\cdot)]+R(s,a)-q(s,a)\right]\right] \\
=& 0.5\mathbb{E}_d[q^2(s,a)] + \mathbb{E}_{(s,a)\sim d^{\mathcal{D}}_w}\left[\mathbb{E}_{s'\sim P(\cdot|s,a)}[\gamma\max q(s',\cdot)]+R(s,a)-q(s,a)\right] \\
=& 0.5\mathbb{E}_d[q^2(s,a)] + \mathbb{E}_{(s,a)\sim d^{\mathcal{D}}_w,r=R(s,a),s'\sim P(\cdot|s,a)}\left[\gamma\max q(s',\cdot)+r-q(s,a)\right] \\
=& \mathcal{L}(d,q,w).
\end{aligned}
$$

This compeletes the proof. $\qquad\square$

**Lemma 14** (Concentration). *For any fixed $d$, with probability at least $1-\delta$, for any $q\in\mathcal{Q}$, $w\in\mathcal{W}$,*

$$
\left|\mathcal{L}(d,q,w)-\hat{\mathcal{L}}(d,q,w)\right| \le \varepsilon_{\text{stat}}.
$$

*Proof.* The statistical error only comes from $\hat{\mathcal{L}}_{\mathcal{W}}$, as

$$
\begin{aligned}
\left|\mathcal{L}(d,q,w)-\hat{\mathcal{L}}(d,q,w)\right| =& \left|\mathbb{E}_{\mathcal{D}}[\hat{\mathcal{L}}(d,q,w)]-\hat{\mathcal{L}}(d,q,w)\right| && \text{\textcolor{red}{(Lemma 13)}} \\
=& \left|\mathbb{E}_{\mathcal{D}}[\hat{\mathcal{L}}_{\mathcal{W}}(q,w)]-\hat{\mathcal{L}}_{\mathcal{W}}(q,w)\right|. && \text{\textcolor{red}{(Eq. (16))}}
\end{aligned}
$$

Since each entry of $\mathcal{L}_{\mathcal{W}}$ is bounded:

$$
\forall q\in\mathcal{Q}, w\in\mathcal{W}, a\in\mathcal{A}, s'\in\mathcal{S}, \quad \left|w(s,a)\left[\gamma\max q(s',\cdot)+r-q(s,a)\right]\right| \le U_{\mathcal{W}}V_{\max},
$$

we can apply Hoeffding's inequality which yields that, for any $q\in\mathcal{Q}$, $w\in\mathcal{W}$, with probability at least $1-\delta/(|\mathcal{Q}||\mathcal{W}|)$,

$$
\left|\mathbb{E}_{\mathcal{D}}[\hat{\mathcal{L}}_{\mathcal{W}}(q,w)]-\hat{\mathcal{L}}_{\mathcal{W}}(q,w)\right| \le U_{\mathcal{W}}V_{\max}\sqrt{\frac{2\log(2|\mathcal{Q}||\mathcal{W}|/\delta)}{N_{\mathcal{D}}}}.
$$

Finally, we can use union bound, rearranging terms to get that, for any fixed $d$, with probability at least $1-\delta$, for any $q\in\mathcal{Q}$, $w\in\mathcal{W}$,

$$
\left|\mathcal{L}(d,q,w)-\hat{\mathcal{L}}(d,q,w)\right| \le U_{\mathcal{W}}V_{\max}\sqrt{\frac{2\log(2|\mathcal{Q}||\mathcal{W}|/\delta)}{N_{\mathcal{D}}}} = \varepsilon_{\text{stat}}
$$

This compeletes the proof. $\qquad\square$

**Lemma 15.** *If $w$ is non-negative $\nu$-a.e. (e.g., $w\in\mathcal{W}$), for any $q\colon\mathcal{S}\times\mathcal{A}\to[0,V_{\max}]$,*

$$
\mathcal{L}(d,q,w)-\mathcal{L}(d,Q^\star,w) \ge 0.5\langle d,q^2-(Q^\star)^2\rangle + \langle(\gamma P_{\pi_e^\star}-I)d^{\mathcal{D}}_w,q-Q^\star\rangle. \tag{17}
$$

*Proof.* This result simply comes from the definition:

$$
\begin{aligned}
&\mathcal{L}(d, q, w) - \mathcal{L}(d, Q^\star, w) \\
=& 0.5\mathbb{E}_d[q^2(s) - (Q^\star)^2(s)] \\
&+ \mathbb{E}_{\mathcal{D}_w}[\gamma \max q(s', \cdot) + r - q(s, a)] - \mathbb{E}_{\mathcal{D}_w}[\gamma \max Q^\star(s', \cdot) + r - Q^\star(s, a)] \\
=& 0.5\mathbb{E}_d[q^2(s) - (Q^\star)^2(s)] \\
&+ \mathbb{E}_{\mathcal{D}_w}[\gamma \max q(s', \cdot) + r - q(s, a)] - \mathbb{E}_{\mathcal{D}_w}[\gamma Q^\star(s', \pi_e^\star) + r - Q^\star(s, a)] \\
\geq& 0.5\mathbb{E}_d[q^2(s) - (Q^\star)^2(s)] \\
&+ \mathbb{E}_{\mathcal{D}_w}[\gamma q(s', \pi_e^\star) + r - q(s, a)] - \mathbb{E}_{\mathcal{D}_w}[\gamma Q^\star(s', \pi_e^\star) + r - Q^\star(s, a)] \\
=& 0.5\mathbb{E}_d[q^2(s) - (Q^\star)^2(s)] \\
&+ \mathbb{E}_{\mathcal{D}_w}[\gamma(q - Q^\star)(s', \pi_e^\star) - (q - Q^\star)(s, a)] \\
=& 0.5\langle d, q^2 - (Q^\star)^2 \rangle + \langle d_w^{\mathcal{D}}, (\gamma P_{\pi_e^\star} - I)(q - Q^\star) \rangle \qquad \text{(Rewrite the expectation with inner products)} \\
=& 0.5\langle d, q^2 - (Q^\star)^2 \rangle + \langle (\gamma P_{\pi_e^\star} - I)d_w^{\mathcal{D}}, q - Q^\star \rangle. \qquad \text{(conjugate)}
\end{aligned}
$$

This compeletes the proof. $\qquad\square$

**Lemma 16.** *If Assumption 5 holds, with probability at least $1 - \delta$, for any $w \in \mathcal{W}$ and any state-action distribution $d$, we have*

$$
\mathcal{L}(d, \hat{q}, w) - \mathcal{L}(d, Q^\star, w) \leq 2\varepsilon_{\text{stat}}. \tag{18}
$$

*Proof.* We can decompose Eq. (18) as follows,

$$
\begin{aligned}
\mathcal{L}(d, \hat{q}, w) - \mathcal{L}(d, Q^\star, w) =& \underbrace{\mathcal{L}(d, \hat{q}, w) - \hat{\mathcal{L}}(d, \hat{q}, w)}_{(1)} + \underbrace{\hat{\mathcal{L}}(d, \hat{q}, w) - \hat{\mathcal{L}}(d, \hat{q}, \hat{w})}_{(2)} \\
&+ \underbrace{\hat{\mathcal{L}}(d, \hat{q}, \hat{w}) - \hat{\mathcal{L}}(d, Q^\star, \hat{w}(Q^\star))}_{(3)} + \underbrace{\hat{\mathcal{L}}(d, Q^\star, \hat{w}(Q^\star)) - \mathcal{L}(d, Q^\star, \hat{w}(Q^\star))}_{(4)} \\
&+ \underbrace{\mathcal{L}(d, Q^\star, \hat{w}(Q^\star)) - \mathcal{L}(d, Q^\star, w)}_{(5)}
\end{aligned}
$$

where $\hat{w}(q) = \operatorname{argmax}_{w \in \mathcal{W}} \hat{\mathcal{L}}(d, q, w)$. For the terms above, we have that:

- (2) and (3) are non-positive since the optimization process.

- (1) and (4) could be bound by concentration.

- For (5), as Bellman optimality equation holds,

$$
\forall s \in \mathcal{S}, a \in \mathcal{A}, \quad \mathbb{E}_{s' \sim P(\cdot|s,a)}[\gamma \max Q^\star(s', \cdot)] + R(s, a) - Q^\star(s, a) = 0.
$$

We have that

$$
\begin{aligned}
(5) =& \mathcal{L}(d, Q^\star, \hat{w}(Q^\star)) - \mathcal{L}(d, Q^\star, w) \\
=& 0.5\mathbb{E}_d[(Q^\star)^2(s, a)] + \mathbb{E}_{\mathcal{D}_{\hat{w}(Q^\star)}}[\gamma \max q(s', \cdot) + r - q(s, a)] \\
&- \left[0.5\mathbb{E}_d[(Q^\star)^2(s, a)] + \mathbb{E}_{\mathcal{D}_w}[\gamma \max Q^\star(s', \cdot) + r - Q^\star(s, a)]\right] \\
=& \mathbb{E}_{(s,a) \sim d_{\hat{w}(Q^\star)}^{\mathcal{D}}, r=R(s,a), s' \sim P(\cdot|s,a)}[\gamma \max Q^\star(s', \cdot) + r - Q^\star(s, a)] \\
&- \left[\mathbb{E}_{(s,a) \sim d_w^{\mathcal{D}}, r=R(s,a), s' \sim P(\cdot|s,a)}[\gamma \max Q^\star(s', \cdot) + r - Q^\star(s, a)]\right] \\
=& \mathbb{E}_{(s,a) \sim d_{\hat{w}(Q^\star)}^{\mathcal{D}}}[\gamma \mathbb{E}_{s' \sim P(\cdot, s, a)}[\max Q^\star(s', \cdot)] + R(s, a) - Q^\star(s, a)] \\
&- \mathbb{E}_{(s,a) \sim d_w^{\mathcal{D}}}[\gamma \mathbb{E}_{s' \sim P(\cdot, s, a)}[\max Q^\star(s', \cdot)] + R(s, a) - Q^\star(s, a)] \\
=& 0.
\end{aligned}
$$

Thus, we conclude that with probability at least $1 - \delta$,

$$
\begin{aligned}
\mathcal{L}(\hat{q}, w) - \mathcal{L}(Q^\star, w) \leq & \underbrace{\mathcal{L}(\hat{q}, w) - \hat{\mathcal{L}}(\hat{q}, w)}_{(1)} + \underbrace{\hat{\mathcal{L}}(Q^\star, \hat{w}(Q^\star)) - \mathcal{L}(Q^\star, \hat{w}(Q^\star))}_{(4)} \\
\leq & |\mathcal{L}(\hat{q}, w) - \hat{\mathcal{L}}(\hat{q}, w)| + |\hat{\mathcal{L}}(Q^\star, \hat{w}(Q^\star)) - \mathcal{L}(Q^\star, \hat{w}(Q^\star))| \\
\leq & 2\varepsilon_{\text{stat}}. && \text{(Lemma 14)}
\end{aligned}
$$

This compeletes the proof. $\qquad\square$

With lemmas above, it's time to prove Lemma 7.

**Lemma** ($L^2$ error of $\hat{q}$ under $d_c$, restatement Lemma 7). *If Assumptions 2, 3, 5, 6 and 8 hold, with probability at least $1 - \delta$, the estimated $\hat{q}$ from Algorithm 1 satisfies*

$$
\|\hat{q} - Q^\star\|_{d_c,2} \leq 2\sqrt{\varepsilon_{\text{stat}}}.
$$

*Proof.* By Assumption 3, $d^{\mathcal{D}}_{w^\star} = (I - \gamma P_{\pi^\star})^{-1} d_c Q^\star$, and from Lemma 15 we have

$$
\begin{aligned}
\mathcal{L}(d_c, \hat{q}, w^\star) - \mathcal{L}(d_c, Q^\star, w^\star) \geq & 0.5\langle d_c, \hat{q}^2 - (Q^\star)^2 \rangle - \langle (I - \gamma P_{\pi^\star})(I - \gamma P_{\pi^\star})^{-1} d_c Q^\star, (\hat{q} - Q^\star) \rangle \\
= & 0.5\langle d_c, \hat{q}^2 - (Q^\star)^2 \rangle - \langle d_c Q^\star, (\hat{q} - Q^\star) \rangle \\
= & 0.5\langle d_c, \hat{q}^2 - (Q^\star)^2 \rangle - \langle d_c, Q^\star(\hat{q} - Q^\star) \rangle \\
= & 0.5\langle d_c, (\hat{q} - Q^\star)^2 \rangle \\
= & 0.5\|\hat{q} - Q^\star\|^2_{d_c,2}.
\end{aligned}
$$

Together with Lemma 16, with probability at least $1 - \delta$,

$$
0.5\|\hat{q} - Q^\star\|^2_{d_c,2} \leq \mathcal{L}(d_c, \hat{q}, w^\star) - \mathcal{L}(d_c, Q^\star, w^\star) \leq 2\varepsilon_{\text{stat}}.
$$

Rearrange this and we can get

$$
\|\hat{q} - Q^\star\|_{d_c,2} \leq 2\sqrt{\varepsilon_{\text{stat}}}
$$

This compeletes the proof. $\qquad\square$

### D.3 Proof of Lemma 8

**Lemma** (Restatement of Lemma 8). *If Assumptions 4 and 7 hold,*

$$
\langle Q^\star(\cdot, \pi^\star_e) - Q^\star(\cdot, \hat{\pi}), \mu_c \rangle \leq 2U_{\mathcal{B}}\|\hat{q} - Q^\star\|_{d_c,1}.
$$

*Proof.* We can rearrange the above term as

$$
\begin{aligned}
\langle Q^\star(\cdot, \pi^\star_e) - Q^\star(\cdot, \hat{\pi}), \mu_c \rangle = & \langle Q^\star(\cdot, \pi^\star_e) - \hat{q}(\cdot, \pi^\star_e), \mu_c \rangle + \langle \hat{q}(\cdot, \pi^\star_e) - \hat{q}(\cdot, \hat{\pi}), \mu_c \rangle \\
& + \langle \hat{q}(\cdot, \hat{\pi}) - Q^\star(\cdot, \hat{\pi}), \mu_c \rangle \\
\leq & \langle Q^\star(\cdot, \pi^\star_e) - \hat{q}(\cdot, \pi^\star_e), \mu_c \rangle + \langle \hat{q}(\cdot, \hat{\pi}) - Q^\star(\cdot, \hat{\pi}), \mu_c \rangle && \text{(Assumption 4)} \\
\leq & \|Q^\star(\cdot, \pi^\star_e) - \hat{q}(\cdot, \pi^\star_e)\|_{\mu_c,1} + \|\hat{q}(\cdot, \hat{\pi}) - Q^\star(\cdot, \hat{\pi})\|_{\mu_c,1} \\
= & \|Q^\star - \hat{q}\|_{\mu_c \times \pi^\star_e,1} + \|\hat{q} - Q^\star\|_{\mu_c \times \hat{\pi},1} \\
\leq & 2U_{\mathcal{B}}\|Q^\star - \hat{q}\|_{d_c,1}
\end{aligned}
$$

The distribution shift comes from the fact that

$$
\left\|\frac{\mu \times \pi_1}{\mu \times \pi_2}\right\|_\infty = \left\|\frac{\pi_1}{\pi_2}\right\|_\infty,
$$

and shifts from $\pi_c$ to $\pi^\star_e$ and $\hat{\pi}$ are both bound by $U_{\mathcal{B}}$ due to Assumptions 4 and 7. This completes the proof. $\qquad\square$

### D.4 Proof of Theorem 1

**Theorem** (Finite sample guarantee of Algorithm 1, restatement of Theorem 1). *If Assumptions 1, 2, 3, 4, 5, 6, 7 and 8 hold with $\varepsilon_c \geq \frac{4C_c U_{\mathcal{B}}\sqrt{\varepsilon_{\text{stat}}}}{1-\gamma}$, then with probability at least $1 - \delta$, the output $\hat{\pi}$ from Algorithm 1 is near-optimal:*

$$J^\star - J_{\hat{\pi}} \leq \frac{4C_c U_{\mathcal{B}}\sqrt{\varepsilon_{\text{stat}}}}{1-\gamma}.$$

*Proof.* From Lemma 7, we have that with probability at least $1 - \delta$,

$$\|\hat{q} - Q^\star\|_{d_c,1} \leq \|\hat{q} - Q^\star\|_{d_c,2} \leq 2\sqrt{\varepsilon_{\text{stat}}}.$$

Then apply Lemma 8 to bound the weighted advantage,

$$\langle Q^\star(\cdot, \pi_e^\star) - Q^\star(\cdot, \hat{\pi}), \mu_c \rangle \leq 2U_{\mathcal{B}}\|\hat{q} - Q^\star\|_{d_c,1} \leq 4U_{\mathcal{B}}\sqrt{\varepsilon_{\text{stat}}}.$$

Finally, according to Lemma 4, $\hat{\pi}$ is $\frac{4C_c U_{\mathcal{B}}\sqrt{\varepsilon_{\text{stat}}}}{1-\gamma}$ optimal. This completes the proof. $\qquad\square$

