# OpenReview forum: "Offline Reinforcement Learning with Additional Covering Distributions"
_TMLR — Accepted by TMLR_

### Review · Reviewer_6vXX · 2023-08-18

**Summary Of Contributions:**

This study centers around offline reinforcement learning employing general function approximation and introduces a novel algorithm named VOPR. In contrast to prior research, the author incorporates an auxiliary covering distribution that effectively reduces the need for excessive trajectories, thereby facilitating the learning of a nearly optimal policy. Furthermore, the proposed algorithm offers finite-sample guarantees based on relatively lenient conditions, encompassing weak realizability and partial coverage assumptions tied to the selected covering distribution.

**Audience:**

Yes

**Broader Impact Concerns:**

There is no concern about the broader impact.

**Claims And Evidence:**

Yes

**Requested Changes:**

See Weakness.

**Strengths And Weaknesses:**

Strengths:

1. The proposed algorithm provides finite-sample guarantees under weaker conditions compared to prior research.

Weaknesses:

1. The paper's clarity and coherence need improvement, as some parts are challenging to comprehend.

(1) It's unclear why Eq. (3) is equivalent to finding a policy $\pi$ that satisfies Equation 4. The correctness only seems apparent when $\hat{q}=Q^*$.

(2) In Lemma 1, there's a lack of definition for the function $J$ and distribution $\mu_{\pi}$. It remains unclear whether distribution $\mu_{\pi}$ originates from the initial distribution $\mu_0$, the covering distribution $\mu_c$, or the data-collection distribution $\mu_D$.

(3) The notation $\mu_a >> \mu_b$ requires further elaboration.

2. The rationale for considering the covering distribution $\mu_c$ isn't adequately explained and needs more elaboration.

(1) Due to the absence of the function $J$ definition, the clarity of the sub-optimality gap definition is compromised. Consequently, it remains unclear why it's sufficient to seek a policy satisfying Equation (4). The potential influence of states outside the $\mu_c$ distribution on the sub-optimality gap also remains unaddressed.

(2) How does the covering distribution $\mu_c$ impact Algorithm 1? It seems that this distribution solely determines the regularization term. It might be more plausible that the covering distribution would further influence the dataset $D$, possibly involving summation over sub-datasets $d_c$.

(3) According to Section 4, the key advantage of the covering distribution $\mu_c$ arises from Assumption 2. With an appropriate distribution, Assumption 2 can be easily satisfied (given a small $C_D$). However, in prior research, partial concentrability seemed to focus solely on the distribution $d_{\pi^*}$ (the optimal policy starting at the initial distribution $\mu_0$), seemingly independent of the dataset $D$ or $d_c$. Could the author provide deeper insights into this assumption?

---

> ### Author Response · Authors · 2023-08-21
>
> We thank the reviewer for the valuable comments.
>
> * 1.(1) This equivalence can be justified with an infinite amount of data and a proper estimation algorithm, in which case $\hat{q}=Q^\star$ on the support of $d_c$. Clarification will be added in the next version.
>
> * 1.(2) The definition of $J$ could be found in section 2, the preliminaries, paragraph Markov Decision Processes (MDPs), line 7, but it needs to be added to the notation list. $\mu_{\pi}$ is the state margin of $d_{\pi}$ and therefore is originated from $\mu_0$. We would also add its definitions into the notation list and to the first mentioned place for clarification.
>
> * 1.(3) This is mentioned in the notation list in Appendix A, which means absolute continuity. For positive-valued function $d_1$ and $d_2$, $d_1\gg d_2$ means that if $d_2>0$, then $d_1> 0$. We typically apply absolute continuity to data distributions. This concept is interesting if we are considering the population setting (also in case we have an infinite amount of data), and we should introduce the concentration coefficient for the empirical setting to derive a non-asymptotic bound. More detailed definitions will be added in the next version.
>
> * 2.(1)
>     * The sub-optimality of a policy $\hat{\pi}$ can be quantified by $\Delta J(\hat{\pi}) = J(\pi^\star)-J(\pi_{\pi})$ (the difference the of expected returns from $\mu_0$; for clarification of the definition of $J$, see 1.(2)).
>     * The sufficient part is justified by the upper bound from Lemma 2 and Lemma 3; they show that a policy satisfying Eq 4. is (near) optimal with a proper $d_c$. Moreover, the necessary part (also the potential influence) is discussed in section 3.1. A more concrete theorem that would be added to the paper is:
>
>         **Theorem**: Assuming $\hat{q}$, $\hat{\pi}$ and a state-action distribution $d$, such that  (i) $d\gg d_{\pi^*}$ for optimal policy $\pi^\star$; (ii) $\langle Q^\star(\cdot, \pi^\star_e) - Q^\star(\cdot, \hat{\pi}), \mu\rangle=0$ (cf. Eq. 4; $\mu$ is the margin of $d$). There exist MDPs with above properties satisfied such that $J_{\pi^\star}-J_{\hat{\pi}}= \frac{1}{1-\gamma}$.
>
>         An intuition for influence of state outside $\mu_c$ is from the induction proofs (proofs of Lemma 2 and Lemma 3), the induction can no longer continue if the coverage assumption is violated.
>
> * 2.(2) $\mu_c$ is the state margin of $d_c$. Its impact on the dataset is assumed in assumption 2.
>
> * 2.(3) A discussion w.r.t. the classic $d_{\pi^\star}$-type data assumption (namely, single optimal coverage; whereas this discussion is more $Q^\star$ estimation focused) could be found in section 3.1. We agree that our data assumptions are more stringent compared with single optimal coverage (as discussed in limitation paragraph). But all most all sample-complexity guarantees obtained for single optimal coverage require either strong assumption on function classes or only work for some specific MDPs (see the paper for references).
> One of our goals is searching for an algorithm and an upper bound without these strong assumptions while possibly introduce (slightly) stronger assumption on the dataset as well as some auxiliary knowledges. Assumption 2 is a part of reflection of this sacrification.

---

### Review · Reviewer_phJY · 2023-08-25

**Summary Of Contributions:**

Their research has introduced a novel PAC offline RL algorithm under "weak realizability" (not encompassing realizability for all classes) and "optimal concentrability" (the offline distribution covers the distribution induced by the optimal policy) with an additional assumption that a certain distribution (such as the offline distribution) covers all near-optimal potentially non-stationary policies.

Their proposed method is comprised of two parts:
* (1): Learn the optimal Q-function so that we can ensure the L2 error guarantee under the covering distribution denoted by $d_c$.
* (2): Learn the optimal policy by setting a policy class $B$ and taking the best policy under the covering distribution $d_c$.

**Audience:**

Yes

**Broader Impact Concerns:**

Not applied

**Claims And Evidence:**

Yes

**Requested Changes:**

* Page 1: Abstract: I believe you had better explain more about *“strong realizably function classes”* and *“weak realizably function classes”* since this jargon is still not standard. I also feel that you had better mention what is “partial coverage" since this usage might not be immediate for most standard readers.
* Page 2: *“Drawbacks.Some works"*: I guess you need a space.
* Page 2: Intro: I feel you had better mention what types of algorithms you use with the covering distribution in the Intro, as I summarized in this review. Otherwise, it is hard to imagine how you use *the covering distribution* from the current intro (before reading the whole paper).
* Page 2: *“Could be meaningless in some cases, and the actual performance of the learned policy is intractable."*: I do not understand what you mean by intractable. They prove that the learned policy is nearly optimal. Would you elaborate on it more?
* Page 2: Table 1. *“We have removed additional definitions of notations for simplicity and refer the interested reader to the original papers for more detail."*  I am not sure this is a good strategy. Generally speaking, I feel the table should be self-contained as much as possible (I assume TMLR does not have limitations for the length of the paper?).
    * It would be better to mention the meaning of each notation even if formally defining is too much, E.g., I personally don’t know (can't recall) what $C(Q)$ in Table 1 is even if I had read all of the papers before. Moreover, I generally believe you had better mention what each notation $w^*,w^{*}_{\alpha}, v_{\alpha},d_c$ roughly means
    * What is "Additional structure"? I think you had probally better say something in the caption that this column corresponds to additionally imposed assumptions on top of *"weak realizability"* and *"optimal concentrability"*.
    * Uehara et al. (2023) technically comprised of two parts. They have a result without a soft-margin; but behavior regularization. So, it might be worthwhile to include it.
    * It might be better to add Xie et.al 2021.
* Page 3: *“We say that a policy is optimal almost everywhere."* Does this mean $\pi^{*}$ might not maximize the value function at regions whose measure is 0?
* Page 4: Did you explain the meaning of the notation $\|...\|_{1,d}$?
* Page 4: *"With assuming the optimal…."* (before Eq 4), I think there is a certain gap. I guess Eq. (3) where we replace $\hat q$ with $Q^*$ is actually equivalent to $\hat \pi$ in (4).
* Page 5: Lemma 2: I think you need to define $\hat \pi$ in (3) by replacing $\hat q$ with $Q^*$ (as you have done in Lemma 3).
* Page 10: *"Nevertheless, the regularization makes the optimality of the learned policy intractable."* Would you elaborate more on what you mean by intractable?

**Strengths And Weaknesses:**

### Strengthens

Their findings are novel, and I consider their contribution to be significant within the offline reinforcement learning theory community.

* Compared to Zhan et al. (2022), they do not rely on behavior regularization.
* Compared to Uehara et al. (2023), they do not rely on soft margins. Algortihmicwise, (2) in the above is different.
* Compared to most offline theoretical RL works, such as Jiang & Huang (2020) and Zhu et al. (2023), they do not rely on strong realizability.


While they rely on the additional assumption (a certain distribution covers all near-optimal, possibly non-stationary policies) compared to these two works, I think this additional assumption is still reasonable.

I checked the central part of the proof. They appear to be correct.

### Weakness

In terms of contributions, the first part of the algorithm (above 1) appears to be a straightforward adaptation of Uehara et al. (2023). Additionally, certain readers may perceive their additional assumption as potentially stringent. However, from my perspective, their core concept remains intriguing and innovative.

The main weakness of this paper is the presentation style. This paper is written in a rigorous way, and I appreciate it. However, certain sections appear challenging to understand for the majority of readers at first sight.

---

> ### Author Response · Authors · 2023-09-06
>
> We thank the reviewer for the valuable comments. We agree with most points of the reviewer. Changes requested and more elaborations will be added in the next version. There are some responses:
>
> > Page 2: “Could be meaningless in some cases, and
>
> "Intractable" means that the performance of the learned policy is not guaranteed. The theorems from Zhan et al., 2022 only ensure that the learned $\hat{\pi}$ is better than $\pi_{\alpha}^\star$ (not the $\pi^\star$ we want), whether $\hat{\pi}$ is optimal is unknown.
>
> > Page 3: “We say that a policy is optimal almost everywhere."  ...
>
> Yes. This is mainly used for situations like continuous state spaces.
>
> > Page 4: Did you explain the meaning ...
>
> Thank you for pointing this out.  $\lVert x\rVert_{p, \nu}$ stands for the $L^p$ norm weighted by measure $\nu$, i.e., $\lVert x\rVert_{p, \nu}=\sqrt[p]{\int |x|^p d\nu}$. Clarification will be added.
>
> > Page 10: "Nevertheless, the regularization makes the optimality of the learned policy intractable." Would you elaborate more ...
>
> See above.

---

> > ### Comment · Reviewer_phJY · 2023-09-25
> >
> > Thx for your response. One thing.
> >
> > > "Intractable" means that the performance of the learned policy is not guaranteed.*
> >
> > Part 2:  I think this is not accurate. In Cor 1, under the coverage for $\pi^{\star}_{\alpha}$, they show the optimality against $\pi^{\star}$. Would you check it? This argument is similarly applied to the first part with Behavior regularization in Uehara et al. (2023).

---

> ### Author Response · Authors · 2023-10-05
>
> We thank the reviewer for pointing this out.
> We have read Zhan et al., 2022 again but there is a concern raised for their proof of theorem 1.
>
> More concretely, in Lemma 10, the $d_{\alpha}^\star$ is not necessarily the induced distribution of $\pi_{\alpha}^\star$, since
> we can only prove that
> $$\sum_{s\in\mathcal{S}}v^\star_{\alpha}(s)\Bigg((1-\gamma)\mu_0(s)+\gamma\sum_{s^\prime, a^\prime}P(s|s^\prime, a^\prime)d_{\alpha}^\star(s^\prime, a^\prime) - d_{\alpha}^\star(s) \Bigg)=0.$$
>
> With an infinite amount of data, PRO-RL is solving the Eqn. (2) in the original paper with the constraint $$\min_{v\in\mathcal{V}}v(s)\sum_{s\in\mathcal{S}}\Bigg((1-\gamma)\mu_0(s)+\gamma\sum_{s^\prime, a^\prime}P(s|s^\prime, a^\prime)d(s^\prime, a^\prime) - d(s) \Bigg)=0,$$ rather than Eqn. (3).

---

> ### Author Response · Authors · 2023-10-07
>
> Sorry to be confusing. Zhan et al., 2022 uses Lemma 3 to show the consistency of the saddle point and thus solves the above concern. But we still cannot see why Lemma 3 holds. Specifically, how can we derive the first inequality of Eqn. (78)?
>
> A counterexample of Lemma 3 can be constructed as the following.
> One can take $\mathcal{X}=\mathcal{X}^\prime=${$(1, -1), (1, 0)$}, $\mathcal{Y}=${$0, 2$}, $\mathcal{Y}^\prime=${$0$} and $f(x, y)=x_1^2+x_2^2+(x_1+x_2)y$. $\Big(x=(1, -1), y=0\Big)$ is the saddle point for $\min_{x\in\mathcal{X}}\max_{y\in\mathcal{Y}}f(x,y)$ and $\max_{y\in\mathcal{Y}}\min_{x\in\mathcal{X}} f(x,y)$.
> In this case,
> arg$\min_{x\in\mathcal{X}^\prime}$arg$\max_{y\in\mathcal{Y}^\prime}=\Big(x=(1,0), y=0\Big)$.

---

> > ### Comment · Reviewer_phJY · 2023-10-23
> > **Response**
> >
> > I think  Eqn. (78) is true.
> > \begin{align}
> > f(x^*, y^*) \leq f(x, y^*) \leq \max_{y \in Y'} f(x,y), x \in \mathcal{X}'
> > \end{align}
> >
> > The first inequity comes from the definition of $x^*$ and $x \in X' \subset X$, and the second inequality is obvious because $y^* \in Y'$.
> >
> > Regarding an example you mentioned for Lemma 3, I am not sure x = (1,-1), y=0 is a saddle point.
> > \begin{align}
> >   \min_{x \in X}f(x , 0) = 1
> > \end{align}
> > and
> > \begin{align}
> >   \min_{x \in X}f(x , 2) = 2
> > \end{align}
> > Hence, $\arg\max_{y \in Y} \min_{x in X}f(x,y) = 2.$ Not 0. Thus, I think saddle point is x = (-1,1), y=2.
> >
> > If you still believe their statement is accurate, I suggest reaching out to the authors in Zhan et al. and engaging in a discussion with them. Generally, it is not within the reviewer's purview to disprove statements in a separate manuscript.
> >
> > After your discussion, if you conclude that their statement is indeed incorrect, you can retain your assertion and possibly specify the section or aspect of their paper that is erroneous in the appendix. Demonstrating the inaccuracy of published manuscripts also constitutes a valuable contribution.
> >
> > However, if you are persuaded that their paper is accurate, I suggest removing the sentence I mentioned concerning "intractable" and acknowledging that they provide valid guarantees for $\pi^{\star}$. This also applies to the initial portion of Uehara et al. (2023).

---

> ### Author Response · Authors · 2023-10-25
>
> We sincerely thank the reviewer for the advice and time, and apologize for the carelessness of the above statement. We have updated the manuscript and hope that this can solve the problems from the reviewer.

---

### Review · Reviewer_Q8GL · 2023-09-13

**Summary Of Contributions:**

This paper presents an offline RL algorithm with general function approximation and partial data coverage under weak realizability and additional information of a covering distribution, which covers all non-stationary near-optimal policies. The paper also discusses the tradeoff between the covering distribution accuracy and the coverage of the dataset.

**Audience:**

Yes

**Claims And Evidence:**

Yes

**Requested Changes:**

Changes that would strengthen the work:
- It is better to explain the notion of covering distribution in the abstract.
- It seems to be helpful if the necessity of the gap assumption without the covering distribution is established (information-theoretic or even lower bound on prior algorithms).

**Strengths And Weaknesses:**

Strengths:
- The paper introduces a possible inductive bias/side information, which is the covering distribution, and how one can use this information to add to the information covered in the offline dataset.
- The paper focuses on eliminating the strong realizability assumptions (a.k.a. Bellman completeness) in offline RL with general function approximation with additional assumptions.
- The paper is well-motivated, well-written, and thoroughly compares with prior work and puts the contribution in perspective. The introduction is particularly nice and insightful, especially for outsiders of the offline RL theory.

Weaknesses:
- The covering distribution assumption is strong and its relevance to practice is not discussed. What are some practical examples in which this covering distribution is available?
- It is unclear to me that the algorithms whose performance upper bounds worsen with the gap (such as Chen & Jiang 2022), indeed suffer from this restriction. In other words, is there a performance lower bound on these algorithms that depends on the gap?

---

> ### Author Response · Authors · 2023-09-13
>
> We thank the reviewer for the valuable comment. There are some responses:
>
> * We agree that assuming a covering distribution is a bit strong. A practical way to obtain a covering distribution can be truncating the data distribution.
>
> *  The gap assumption made in Chen & Jiang 2022 is somehow special: they require that the optimal action be unique in each state. An algorithm-specific counterexample for Chen & Jiang 2022 can be found in Figure 1 in their paper. The MDP in Figure 1 is not solvable by their algorithm.

---

### Public Comment · ~Nan_Jiang2 · 2023-11-30
**Neat idea**

Just come across this paper today and it looks pretty neat! Will give a closer read later.

One comment: assuming I understand the main message correctly, I believe it's worth highlighting that you only need "marginal" data (s,a) for the d_c distribution that is required to provide (some form of) all-policy coverage. So one can construct a valid d_c distribution even without knowledge of the dynamics (since (r,s') is not required). We have a paragraph describing this point in Huang & Jiang'22 (https://arxiv.org/pdf/2210.15543.pdf, paragraph below Section 4.1).

---

> ### Author Response · Authors · 2023-12-11
> **Thank you for your comments and reminder**
>
> We agree with you and the point that one can "model-freely" construct $d_c$ is worth a highlight. Personally, we believe that the question of how much we can do without the knowledge of MDPs is one of the central topics in offline RL, and we are excited to investigate it.
> Thank you again for raising this point, we shall dig into Huang & Jiang'22 https://arxiv.org/pdf/2210.15543.pdf later.

---

### Decision · Action_Editor_P9VK · 2023-11-02

**Recommendation:** Accept as is

**Comment:**

This paper proposes a new analysis of offline RL under weak realizability and optimal concentrability with the new assumption of accessing an additional covering distribution. The reviewers are in consensus that this work relaxes the assumptions in many previous works e.g. Bellman completeness, soft gap assumptions, which is a nice contribution. They agree that this paper should be accepted to TMLR.

**Audience:**

Yes

**Claims And Evidence:**

Yes